# KnowGPT: Knowledge Graph based PrompTing for Large Language Models

**Qinggang Zhang**[*†]**, Junnan Dong**[*†]**, Hao Chen**[†]**, Daochen Zha**[‡]**, Zailiang Yu**[§]**, Xiao Huang**[†]

[†]The Hong Kong Polytechnic University, [‡]Rice University, [§]Zhejiang Lab

{qinggangg.zhang,hanson.dong}@connect.polyu.hk, sundaychenhao@gmail.com
daochen.zha@rice.edu, yuzl@zhejianglab.com,xiaohuang@comp.polyu.edu.hk

## Abstract

Large Language Models (LLMs) have demonstrated remarkable capabilities in many real-world applications. Nonetheless, LLMs are often criticized for their tendency to produce hallucinations, wherein the models fabricate incorrect statements on tasks beyond their knowledge and perception. To alleviate this issue, graph retrieval-augmented generation (GraphRAG) has been extensively explored which leverages the factual knowledge in knowledge graphs (KGs) to ground the LLM's responses in established facts and principles. However, most state-of-the-art LLMs are closed-source, making it challenging to develop a prompting framework that can efficiently and effectively integrate KGs into LLMs with hard prompts only. Generally, it usually suffers from three critical issues, including huge search space, high API costs, and laborious prompt engineering, that impede the widespread application in practice. To this end, we introduce a novel **Know**ledge **G**raph-based **P**romp**T**ing framework, namely KnowGPT, to enhance LLMs with domain knowledge. KnowGPT contains a knowledge extraction module to extract the most informative knowledge from KGs, and a context-aware prompt construction module to automatically convert extracted knowledge into effective prompts. Experiments on three benchmark datasets demonstrate that KnowGPT significantly outperforms all competitors including the state-of-the-art GraphRAG models. Notably, KnowGPT achieves a 92.6% accuracy on OpenbookQA leaderboard, close to human-level performance.

## 1 Introduction

Large Language Models (LLMs), such as GPT-4 [54] and Claude 3 [3], have surprised the world with superior performance in a wide range of real-world applications [14, 38, 82, 83]. Despite their impressive performance, LLMs are frequently criticized for their limited ability to handle factual information accurately and their tendency to generate hallucinations [18], especially when faced with questions requiring domain-specific or professional knowledge not covered in their training corpus [2, 59]. For example, when queried about nutrient composition, an LLM might erroneously associate it with "energy", as depicted in Figure 1. This error stems from the model's insufficient biological knowledge. Therefore, integrating domain knowledge into LLMs is crucial for reducing hallucinations and unlocking their full potential in diverse industry applications [24].

Recently, retrieval-augmented generation (RAG) has been explored, which can enhance LLMs with external knowledge from text corpora or online sources [41, 89, 53]. It combines LLMs with external knowledge retrieval systems to help reduce hallucinations. However, these models face challenges in real-world applications due to the varying quality of available data. Domain knowledge is often scattered across different sources, such as textbooks, research papers, technical manuals, and industry

---

[*]Equal contribution

38th Conference on Neural Information Processing Systems (NeurIPS 2024).

reports [42]. These textual documents may have varying levels of quality, accuracy, and completeness, leading to potential inconsistencies or errors in the retrieved knowledge [92].

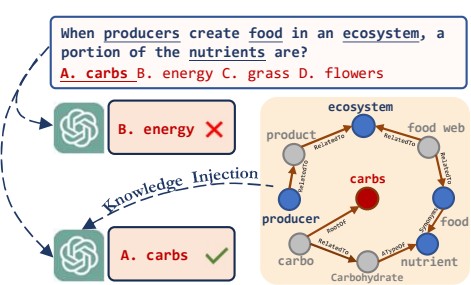

Figure 1: A real-world question from Open-bookQA. GPT-3.5 could effectively correct the answer given the scientific reasoning background from ConceptNet (blue: question concepts, red: answers, grey: entities not present in questions).

A promising avenue for addressing the above issue entails the integration of Knowledge Graphs (KGs) into LLMs. KGs provide a structured representation of domain knowledge, as they are constructed based on rigid ontologies that clearly define the jargon, acronyms, specialized terminologies and their relationships in specific domains [43, 60, 84, 85, 86]. The enormous factual knowledge stored in KGs holds the potential to ground the model's responses in established facts and principles [29, 77, 56]. For instance, in Figure 1, an LLM can correct itself by leveraging the related background knowledge in ConceptNet [62]. Earlier studies adopted a heuristic way to inject knowledge from KGs into the LLMs during pre-training or fine-tuning. For example, ERNIE [63] incorporates entity embeddings and aligns them with word embeddings in the pre-training phase, encouraging the model

to better understand and reason over entities. Similarly, KnowBERT [57] integrates entity linkers with BERT to infuse knowledge about entities during fine-tuning for knowledge-intensive applications. Another line of work focuses on retrieving relevant knowledge from KGs at inference time to augment the language model's context. Typically, K-BERT [47] uses an attention mechanism to select relevant triples from KGs based on the query context, which are then appended to the input sequence. Similarly, KEPLER [73] learns joint embeddings of text and KG entities to enhance the model's predictions. Subsequent works have further integrated graph neural networks alongside LLMs for joint reasoning [80, 87, 15] and introduced interactions between text tokens and KG entities within the intermediate layers of LLMs [64, 67].

However, as LLMs have been keeping evolving, most SOTA LLMs remain *closed-source* in practice. For instance, GPT-4 [54] and Claude 3 [3] exclusively grant access through their APIs, which means we can only retrieve model responses by submitting textual inputs, with model specifics inaccessible. As such, the research focus has recently shifted towards KG prompting that enhances fixed LLMs with KG-based hard prompts [46, 56]. KG Prompting for LLMs has been a new learning paradigm in natural language processing. Specifically, CoK [71] introduces Chain-of-Knowledge prompting to decompose LLM-generated reasoning chains into evidence triples, verifying their factuality and faithfulness using an external KG. Mindmap [74] provides more transparency on LLMs' decision-making by enabling comprehension and reasoning over structured KG inputs. RoG [50] presents a planning-retrieval-reasoning framework that synergizes LLMs and KGs for more transparent and interpretable reasoning. KGR [25] proposes an autonomous approach to retrofit LLM-generated responses by leveraging KGs to extract, verify, and refine factual information throughout the reasoning process, effectively mitigating hallucination for knowledge-intensive applications.

Despite the promising performance of existing KG prompting methods, three critical issues hinder their widespread application in practice. ❶ Huge search space. Real-world KGs often consist of millions of triples, resulting in a vast search space when retrieving relevant knowledge for prompting. ❷ High API cost. Closed-source LLMs, like GPT-4 and Claude 3, are accessible through proprietary APIs, which can incur significant costs when performing KG prompting at scale [17]. Thus, careful selection of the most informative knowledge from KGs is essential to minimize costs. ❸ Laborious prompt design. LLMs are highly sensitive to prompts, with even minor variations in prompts conveying the same semantic meaning potentially yielding drastically different responses. However, existing methods rely on manually designed or rule-based prompts to present factual knowledge from KGs. These hard prompts are inherently inflexible and rigid, lacking the adaptability to accommodate variations in question semantics and KG structures.

To this end, we propose a novel **Know**ledge **G**raph-based **P**romp**T**ing framework, namely **KnowGPT**, which leverages the factual knowledge in KGs to ground the model's responses in established facts and principles. In this paper, we aim to answer two key research questions. ❶ Given a query and a large-scale KG, how could we effectively and efficiently retrieve factual knowledge from KG that is

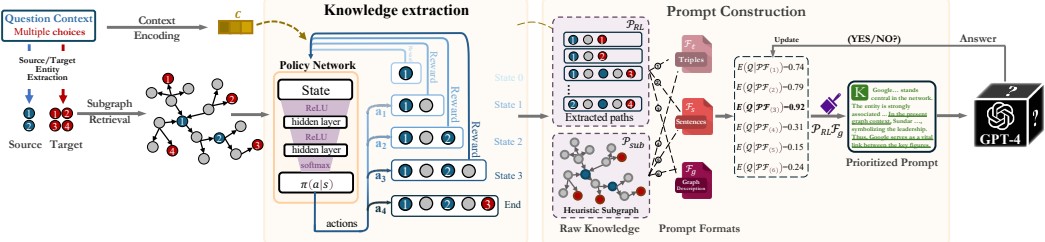

Figure 2: The overall architecture of our proposed knowledge graph prompting framework, i.e., KnowGPT. Given the question context with multiple choices, we first retrieve a question-specific subgraph from the real-world KG. *Knowledge Extraction* is first dedicated to searching for the most informative and concise reasoning background subject to the context. Then the *Prompt Construction* module is optimized to prioritize the combination of knowledge and formats subject to the given question.

relevant to the query? ❷ Given the raw knowledge extracted from KGs, how could we convert the extracted knowledge into an effective prompt that is easily understandable for LLM?

We shed light on the the above questions with a novel prompt learning method that is rather effective, generalizable, and cost-efficient. Specifically, to address question ❶, we leverage deep reinforcement learning (RL) to extract the most informative knowledge from KGs. To encourage the agent to discover more informative knowledge chains, we devise a tailored reward scheme that promotes the reachability, context-relatedness, and conciseness of the extracted paths. Then, a policy network is trained to maximize the reward using training questions and applied to unseen questions. To tackle question ❷, we introduce a prompt construction strategy based on Multi-Armed Bandit (MAB). Given several knowledge extraction strategies and prompt templates, an MAB is learned to select the most effective combination for each question by balancing exploration and exploitation. The learned MAB is then applied to new questions to select knowledge extraction strategies and prompt templates automatically. Our main contributions are summarized as follows:

- Formally define the problem of KG-based prompting, which leverages the structured knowledge in KGs to ground the LLM's responses in established facts and principles.
- Propose KnowGPT, a novel prompting framework that leverages deep reinforcement learning (RL) and Multi-Armed Bandit (MAB) to generate effective prompts for domain-specific queries.
- Implement KnowGPT upon GPT-3.5. Experiments on three QA datasets shows KnowGPT outperforms SOTA baseline models by a large margin. Notably, KnowGPT achieves an average improvement of 23.7% over GPT-3.5 and an average improvement of 2.9% over GPT-4. Additionally, it attains a 92.6% accuracy on the OpenbookQA leaderboard, which is comparable to human performance.

## 2 Problem Statement

We formally define the problem of *knowledge graph based prompting for LLMs* in question answering. We represent each question as a question context $\mathcal{Q} = \{\mathcal{Q}_s, \mathcal{Q}_t\}$, where $\mathcal{Q}_s = \{e_1, e_2, ..., e_m\}$ is a set of $m$ *source entities*, and $\mathcal{Q}_t = \{e_1, e_2, ..., e_n\}$ is a set of $n$ *target entities*. Following prior work [20, 79], $\mathcal{Q}_s$ is extracted by concept recognition, and we assume it is given in our problem. Similarly, each target entity in $\mathcal{Q}_t$ is extracted from a corresponding candidate answer. We denote an LLM as $f$, a real-world KG as $\mathcal{G}$, which consists of triples (head entity, relation, tail entity), denoted as $(h, r, t)$. In our setting, we only have access to the APIs of $f$. However, we can employ open-source lightweight language models (not $f$), like Bert-Base [35], to initialize question embeddings. Using the above notations, we describe our problem below.

> Given a question $\mathcal{Q}$, an LLM $f$, and a domain KG $\mathcal{G}$, we aim to learn a prompting function $f_{\text{prompt}}(\mathcal{Q}, \mathcal{G})$, which generates a prompt $\mathbf{x}$ that incorporates the context of $\mathcal{Q}$ and the factual knowledge in $\mathcal{G}$, such that the prediction of the LLM $f(\mathbf{x})$ can output the correct answers for $\mathcal{Q}$.

## 3 KnowGPT Framework

Learning the prompting function $f_{\text{prompt}}(\mathcal{Q}, \mathcal{G})$ involves two challenges, i.e., what knowledge should be used in $\mathcal{G}$, and how to construct the prompt. To address these challenges, we present KnowGPT,

which extracts raw knowledge with deep RL and then constructs the prompt with MAB. An overview of our framework is shown in Figure 2.

## 3.1 Knowledge Extraction with Deep Reinforcement Learning

Intuitively, the relevant reasoning background lies in a question-specific subgraph $\mathcal{G}_{sub}$ that contains all the *source* entities $\mathcal{Q}_s$, *target* entities $\mathcal{Q}_t$, and their neighbors. An ideal subgraph $\mathcal{G}_{sub}$ is expected to have the following properties: $(i)$ $\mathcal{G}_{sub}$ encompasses as many source and target entities as possible, $(ii)$ the entities and relations within $\mathcal{G}_{sub}$ exhibit a strong relevance to question context, and $(iii)$ $\mathcal{G}_{sub}$ is concise with little redundant information such that it can be fed into LLMs with limited lengths.

However, it is challenging to find such a $\mathcal{G}_{sub}$ since extracting a subgraph is NP-hard. To effectively and efficiently find a satisfactory $\mathcal{G}_{sub}$, we develop a tailored knowledge extraction method, named $\mathcal{P}_{RL}$, that employs deep RL to sample reasoning chains in a trial-and-error fashion. Specifically, we assume $\mathcal{G}_{sub}$ is constructed based on a set of reasoning chains $\mathcal{P} = \{\mathcal{P}_1, \mathcal{P}_2, ..., \mathcal{P}_m\}$, where each knowledge chain $\mathcal{P}_i = \{(e_i, r_1, t_1), (t_1, r_2, t_2), ..., (t_{|\mathcal{P}_i|-1}, r_{|\mathcal{P}_i|}, t_{|\mathcal{P}_i|})\}$ is a path in $\mathcal{G}$ starting from the $i\text{-}th$ source entity in $\mathcal{Q}_s$, and $|\mathcal{P}_i|$ is the path length. $\mathcal{G}_{sub}$ encompasses all the entities and relations appeared in $\mathcal{P}$.

- **State:** A state indicates the current location in KG, i.e., one of the entities in KG. Specifically, it represents the spatial change from entity $h$ to $t$. Inspired by the prior study [76], we define the state vector $s$ as:
$$s_t = (e_t, e_{target} - e_t), \tag{1}$$
where $e_t$ and $e_{target}$ are the embedding vectors of the current entity and the target entity. To get the initial node embeddings for entities extracted from the background KG, we adopt the approach proposed by the previous study [20]. Specifically, we transform knowledge triples from the KG into sentences and feed them into pre-trained LM to get node embeddings.

- **Action:** The action space encompasses all the neighboring entities of the current entity, enabling the agent to explore the KG flexibly. By taking an action, the agent will move from the current entity to the chosen neighboring entity.

- **Transition:** The transition model P measures the probability of moving to a new state $(s')$ given existing state $(s)$ and the undertaken action $(a)$. In KGs, the transition model takes on the form $P(s'|s, a) = 1$ if $s$ is directed to $s'$ through action $a$; Otherwise, $P(s'|s, a) = 0$.

- **Reward:** To determine the quality of the formed path, we define the reward based on reachability:
$$r_{reach} = \begin{cases} +1, & if\ target; \\ -1, & otherwise, \end{cases} \tag{2}$$
which represents whether the path eventually reaches the target within limited steps. Specifically, the agent receives a reward of $+1$ if it can attain the target within $K$ actions. Otherwise, it will receive $-1$ as the reward.

Reaching a target entity is not our sole focus. To avoid overlong and rigmarole reasoning chains, we also design two auxiliary rewards to promote context-relatedness and path conciseness.

### 3.1.1 Context-relatedness Auxiliary Reward

The key motivation is to encourage paths closely related to the given question context. Specifically, we evaluate the semantic relevance of a path $\mathcal{P}_i$ to the context $\mathcal{Q}$. Inspired by the prevailing study [80], a fixed but well-trained matrix $W$ is applied to map the path embedding $\mathcal{P}$ to the same semantic space with context embedding $c$. To this end, this auxiliary reward is formulated as:
$$r_{cr} = \frac{1}{|i|} \sum_{source}^{i} cos(W \times \mathcal{P}_i, c), \tag{3}$$
where $c$ is the embedding of context $\mathcal{Q}$ we obtained from a pre-trained LM [35] and the embedding of path $\mathcal{P}_i$ is the average of the embeddings of all the entities and relations we have walked through till $i$, i.e., $Avg(e_{source} + re_1... + e_i)$, where $i \leq length(\mathcal{P}_{target})$. This step-by-step reward scheme provides rewards before the target is reached.

### 3.1.2 Conciseness Auxiliary Reward

There are two additional significant challenges for the candidate reasoning background. $(i)$ The natural limitation of LLMs for over-long context understanding gives constrained budgets for prompts, where the extracted knowledge chain is expected to be concise enough to ensure the full understanding by closed-source LLMs. $(ii)$ The prohibitive cost of calling LLMs' API guides the prompt to be more concise. By limiting the step size, we encourage the policy to find as much valuable information as possible within the shortest path length.

Considering the inevitable homogeneity in the large-scale real-world KG constructed from the online corpus, each step in the final path is ideally a necessity. Specifically, we evaluate the conciseness of a path to reduce twists and turns on redundant entities, e.g., synonyms. Thus, the reward for the conciseness of a path $\mathcal{P}_i$ is formulated as follows.

$$r_{\text{cs}} = \frac{1}{|\mathcal{P}_i|}. \tag{4}$$

To this end, our overall reward modeling consists of three major criteria that comprehensively incentivize the entire policy learning for an effective knowledge extraction.

### 3.1.3 Training Policy Network

To solve the MDP defined above, a tailored policy network $\pi_\theta(s, a) = p(a|s; \theta)$ is trained to extract a reasoning chain in the KG. We optimize the network with policy gradient [76]. The optimal policy navigates the agent from the source entity to the target entity while maximizing the accumulated rewards. We provide more training details in the Appendix.

## 3.2 Prompt Construction with Multi-armed Bandit

In this subsection, we design a tailored prompt construction strategy based on Multi-Armed Bandit (MAB). The key idea is to learn to select the best knowledge extraction and prompt templates at a meta-level. We will begin by outlining the overall strategy, followed by detailing its instantiation with two knowledge extraction methodologies and three templates.

Suppose we have several knowledge extraction strategies $\{\mathcal{P}_1, \mathcal{P}_2, ..., \mathcal{P}_m\}$ and several candidate prompt formats $\mathcal{F} = \{\mathcal{F}_1, \mathcal{F}_2, ..., \mathcal{F}_n\}$. Each knowledge extraction strategy $\mathcal{P}_i$ is a method for selecting reasoning background given a question context, such as the RL-based strategy discussed above. Every prompt template $\mathcal{F}_j$ represents a mechanism to transform the triples within the subgraph into a prompt for an LLM prediction.

The prompt construction problem is to identify the best combination of $\mathcal{P}$ and $\mathcal{F}$ for a given question. We define the overall process of selection as a reward maximization problem, $\max \sum r_{pf}$, where $r_{pf}$ is obtained as:

$$\sigma(f(\mathcal{PF}_{(i)})) = \begin{cases} 1 & if\ accurate; \\ 0 & otherwise. \end{cases} \tag{5}$$

Specifically, $\mathcal{PF}_{(i)}, i \in \{0, 1, \cdots, m \times n\}$ is one of the combination, and $r_{pf} \in \{0, 1\}$ indicates the performance of the output of LLM in answering the current question.

To capture the context-aware correlation between questions and different combinations of knowledge and prompt formats, we formulate the selection mechanism of MAB with an expectation function $E(\cdot)$. It adaptively measures the potential expectation of a combination for different questions.

$$E(\mathcal{Q}|\mathcal{PF}_{(i)}) = \boldsymbol{c} \times \boldsymbol{\alpha}_{(i)} + \beta_{(i)}. \tag{6}$$

Here, $\boldsymbol{c}$ represents the embedding of $\mathcal{Q}$. The vector $\boldsymbol{\alpha}(i)$ corresponds to a set of non-negative parameters associated with $\mathcal{PF}(i)$, which have been learned during the previous $k$-1 iterations. Additionally, $\beta_{(i)}$ stands for a balancing factor introducing noise according to a Gaussian distribution.

Empirically maximizing $\boldsymbol{c} \times \boldsymbol{\alpha}_i$ could encourage exploitation [12, 16] for the best combination, we could effectively update $\boldsymbol{\alpha}_{(i)}$ via modeling the correlations between the context embedding of the anchor question $\mathbf{c}_i$ and all the previously selected contexts $\mathbf{C}_{(i)}$ for particular combination

$\mathcal{PF}_{(i)}$ in former $k$ steps, and the rewards $\mathbf{r}_{pf}^{(i)}$ obtained from the selection of the current combination. Concretely, the $\boldsymbol{\beta}^{(b)}$ is updated as:

$$J(\mathbf{C}_{(i)}^{(k)}, \mathbf{r}_{pf}^{(i)(k)}) = \sum_{k=1}^{K}(\mathbf{r}_{pf}^{(i)(k)} - \mathbf{C}_{(i)}^{(k)}\boldsymbol{\alpha}^{(i)})^2 + \lambda^i \parallel \boldsymbol{\alpha}^{(i)} \parallel_2^2 .$$

$$\rightarrow \boldsymbol{\alpha}^{(i)} = \left((\mathbf{C}_{(i)}^{(k)})^\top \mathbf{C}_{(i)}^{(k)} + \lambda^i \mathbf{I}\right)^{-1} (\mathbf{C}_{(i)}^{(k)})^\top \mathbf{r}_{pf}^{(i)(k)}. \tag{7}$$

Here, $J$ denotes the OLS training loss. $\mathbf{I} \in \mathbb{R}^{d \times d}$ is an identity matrix and $\lambda^i$ is a regularization factor that controls the complexity of the model.

Similarly, in order to encourage exploration within less frequently selected pairings, we employ an upper confidence bound approach to balance exploration and exploitation. This is achieved through the introduction of the parameter $\beta^{(i)}$. Inspired by prevailing studies [70, 16], we can derive the following exploration term $\beta^{(i)}$:

$$\beta^{(i)} = \gamma \times \sqrt{\mathbf{c}_i \left((\mathbf{C}_{(i)}^{(k)})^\top \mathbf{C}_{(i)}^{(k)} + \lambda^i \mathbf{I}\right)^{-1} (\mathbf{c}_{(i)})^\top}, \tag{8}$$

where $\gamma$ is a fixed constant, i.e., $\gamma = 1 + \sqrt{ln(2/\delta)/2}$.

When the model picks a combination with a large $\boldsymbol{c} \times \boldsymbol{\alpha}_i$, it signifies an exploitation process. Likewise, when the model selects a combination with larger $\beta^{(i)}$, this variance indicates an exploration process due to the model making fewer selections of the current combination. Thus, jointly maximizing $\boldsymbol{c} \times \boldsymbol{\alpha}_i + \beta_{(i)}$ could help us get rid of the dilemma of exploration and exploitation.

Consequently, our MAB design can leverage the feedback from the LLM to optimize the selection policy. By maximizing the expectation function $E(\cdot)$, it learns to balance the exploitation and exploration to prioritize the most promising prompts for specific question contexts.

### 3.2.1 Implementation

We implement the above MAB strategies with two knowledge extraction strategies and three templates. Note that our MAB design is general and can be implemented with more knowledge extraction strategies and prompt templates for better performance. The knowledge extraction strategies include:

- $\mathcal{P}_{\text{RL}}$: The RL-based knowledge extraction strategy presented in the previous subsection.
- $\mathcal{P}_{\text{sub}}$: A heuristic sub-graph extraction strategy that extracts a 2-hop subgraph around both the source and target entities. Detailed implementation can be found in Section B.1 of Appendix. Since RL is notoriously unstable [65], we introduce $\mathcal{P}_{\text{sub}}$ as an alternative candidate strategy for the MAB selection, ensuring a fallback option if the RL-based approach does not perform well.

The prompt templates include:

- **Triples**, denoted as $\mathcal{F}_t$, are indeed the originally extracted knowledge and empirically tested that could be understood by the black-box LLMs, e.g., (*Sergey_Brin*, *founder_of*, *Google*),(*Sundar_Pichai*, *ceo_of*, *Google*), (*Google*, *is_a*, *High-tech Company*).
- **Sentences** is a following solution to transform the knowledge into a colloquial $\mathcal{F}_s$, e.g., '*Sergey Brin, who is a founder of Google, a high-tech company, has now passed the reigns to Sundar Pichai, who is currently serving as the CEO of the company.*'
- **Graph Description**, $\mathcal{F}_g$ prompts the LLM by treating the knowledge as a structured graph. We preprocess the extracted knowledge with black-box LLM itself to generate the description by highlighting the center entity, e.g., '*Google, a high-tech company, stands central in the network. The entity is strongly associated with significant individuals in the tech industry. Sergey Brin, one of the founders, established Google, underscoring its historical beginnings. In the present graph context, Sundar Pichai is recognized as the CEO of Google, symbolizing the company's current leadership. Thus, Google serves as a vital link between these key figures.*'

Considering two knowledge extraction methods: $\mathcal{P}_{\text{sub}}$ and $\mathcal{P}_{\text{RL}}$, as well as three prompt translation methods: $\mathcal{F}_t$, $\mathcal{F}_s$ and $\mathcal{F}_g$, the MAB is trained to learn from the feedback from LLMs

to prioritize the most appropriate combination among two extraction methods and three pre-defined prompt formats for different real-world question contexts, i.e., $\mathcal{PF} = \{(\mathcal{P}_{sub}\mathcal{F}_t), (\mathcal{P}_{sub}\mathcal{F}_s), (\mathcal{P}_{sub}\mathcal{F}_g), (\mathcal{P}_{RL}\mathcal{F}_t), (\mathcal{P}_{RL}\mathcal{F}_s), (\mathcal{P}_{RL}\mathcal{F}_g)\}$.

# 4 Experiments

We conduct extensive experiments to evaluate KnowGPT on three benchmark question-answering datasets, covering both commonsense and domain-specific QA. We implement KnowGPT upon GPT-3.5. Our experiments are designed to answer the following research questions:

- **RQ1 (Main results)**: How does KnowGPT perform when compared with the state-of-the-art LLMs and KG-enhanced QA baselines?

- **RQ2 (Ablation Study)**: How does each key component of KnowGPT contribute to the performance?

- **RQ3 (Case study)**: How could KG help solve complex reasoning tasks? See Appendix 4.4.

## 4.1 Experimental Setup

**Datasets**. We evaluate KnowGPT on three QA datasets spanning two fields: CommonsenseQA [66] and OpenBookQA [52] serve as benchmarks for commonsense reasoning, while MedQA-USMLE [34] acts as a domain-specific QA benchmark. The statistics of these three datasets can be found in Table 5 in the Appendix.

**Baselines**. We carefully select baseline models from four categories for a comprehensive evaluation.

*LM + Fine-tuning*. We compare our method with vanilla fine-tuned LMs. Specifically, we choose Bert-base, Bert-large [35], and RoBerta-large [49] as representative fine-tune LM methods. To conduct commonsense and biomedical QA, we fine-tune these three LMs via additional linear layers.

*KG-enhanced LM*. We have also implemented several recently released models for integrating KGs into question answering, which encompass MHGRN [20], QA-GNN [80], HamQA [15], JointLK [64], GreaseLM [88] and GrapeQA [67]. To ensure a fair comparison, we implement these baselines with advanced language models that are optimized for particular datasets. Specifically, RoBerta-large [49] is used for CommenseQA, while AristoRoBERTa [13] is designated for OpenBookQA. For MedQA, we opt for the top-tier biomedical language model, SapBERT [45]. Note that due to the white-box nature of these methods and their high computation overheads, it is infeasible to apply them to state-of-the-art LLMs, like GPT-3.5 and GPT-4.

*LLM + Zero-shot*. We include several representative generative LLMs, including ChatGLM, ChatGLM2, Baichuan-7B, InternLM, GPT-3, GPT-3.5 and GPT-4 as knowledge-agnostic alternatives. Specifically, we used the model 'text-davinci-002' provided by OpenAI as the implementation of GPT-3, and 'gpt-3.5-turbo' and 'gpt-4' as the implementations of GPT-3.5 and GPT-4, respectively (we have provided more implementation details of all LLMs in Appendix A.4). The question-answering task is conducted under the zero-shot setting with the question query from the test set as input.

*LLM + KG Prompting*. To verify the effectiveness of our prompting strategy, we also add the state-of-the-art KG prompting methods, i.e., CoK [71], RoG [50], and Mindmap [74] as baselines. Notably, we did include KGR [25] in our main results, since the authors have not released their codes.

## 4.2 Main Results (RQ1)

To address **RQ1**, we evaluate KnowGPT by comparing it to state-of-the-art baselines on the three benchmark datasets. KnowGPT is based on the original GPT-3.5. We measure the performance using accuracy, which calculates the percentage of questions correctly predicted by the model out of the total questions in the test set. We have the following observations:

- KnowGPT outperforms all categories of methods, including sixteen different baselines, across all datasets and model architectures. This suggests that KnowGPT can effectively inject the knowledge from KGs to LLMs.

- KnowGPT surpasses the performance of GPT-3.5 and even GPT-4. On average, KnowGPT achieves a 23.7% higher testing accuracy than GPT-3.5. Specifically, KnowGPT outperforms GPT-3.5 by 10.8%,

Table 1: Performance comparison among baseline models and KnowGPT on three benchmark datasets.

| Catagory | Model | CommonsenseQA | | OpenBookQA | | MedQA | |
|---|---|---|---|---|---|---|---|
| | | IHdev-Acc. | IHtest-Acc. | Dev-Acc. | Test-Acc. | Dev-Acc. | Test-Acc. |
| LM + Fine-tuning | Bert-base | 0.573 | 0.535 | 0.588 | 0.566 | 0.359 | 0.344 |
| | Bert-large | 0.611 | 0.554 | 0.626 | 0.602 | 0.373 | 0.367 |
| | RoBerta-large | 0.731 | 0.687 | 0.668 | 0.648 | 0.369 | 0.361 |
| KG-enhanced LM | MHGRN | 0.745 | 0.713 | 0.786 | 0.806 | - | - |
| | QA-GNN | 0.765 | 0.733 | 0.836 | 0.828 | 0.394 | 0.381 |
| | HamQA | 0.769 | 0.739 | 0.858 | 0.846 | 0.396 | 0.385 |
| | JointLK | 0.777 | 0.744 | 0.864 | 0.856 | 0.411 | 0.403 |
| | GreaseLM | 0.785 | 0.742 | 0.857 | 0.848 | 0.400 | 0.385 |
| | GrapeQA | 0.782 | 0.749 | 0.849 | 0.824 | 0.401 | 0.395 |
| LLM + Zero-shot | ChatGLM | 0.473 | 0.469 | 0.352 | 0.360 | 0.346 | 0.366 |
| | ChatGLM2 | 0.440 | 0.425 | 0.392 | 0.386 | 0.432 | 0.422 |
| | Baichuan-7B | 0.491 | 0.476 | 0.411 | 0.395 | 0.334 | 0.319 |
| | InternLM | 0.477 | 0.454 | 0.376 | 0.406 | 0.325 | 0.348 |
| | Llama2 (7b) | 0.564 | 0.546 | 0.524 | 0.467 | 0.338 | 0.340 |
| | Llama3 (8b) | 0.745 | 0.723 | 0.771 | 0.730 | 0.639 | 0.697 |
| | GPT-3 | 0.539 | 0.520 | 0.420 | 0.482 | 0.312 | 0.289 |
| | GPT-3.5 | 0.735 | 0.710 | 0.598 | 0.600 | 0.484 | 0.487 |
| | GPT-4 | 0.776 | 0.786 | 0.878 | 0.910 | 0.739 | 0.763 |
| LLM + KG Prompting | CoK | 0.759 | 0.739 | 0.835 | 0.869 | 0.706 | 0.722 |
| | RoG | 0.750 | 0.734 | 0.823 | 0.861 | 0.713 | 0.726 |
| | Mindmap | 0.789 | 0.784 | 0.851 | 0.882 | 0.747 | 0.751 |
| Ours | KnowGPT | **0.827** | **0.818** | **0.900** | **0.924** | **0.776** | **0.781** |
| KnowGPT vs. GPT-3.5 | + 23.7% (Avg.) | + 9.2% | + 10.8% | + 31.2% | + 32.4% | + 29.2% | + 29.4% |
| KnowGPT vs. GPT-4 | +2.9% (Avg.) | + 5.1% | + 3.3% | + 2.2% | + 1.4% | + 3.7% | + 1.8% |

*We used 'text-davinci-002' and 'gpt-3.5-turbo' provided by OpenAI as the implementation of GPT models.
*The results compared with fine-tuning LLMs on CommonsenseQA are placed in Table 6 of Appendix.

32.4%, and 29.4% on the CommonsenseQA, OpenBookQA, and MedQA datasets, respectively. More importantly, despite being based on GPT-3.5, KnowGPT outperforms the state-of-the-art LLM GPT-4 by 3.3%, 1.4%, and 1.8% on the CommonsenseQA, OpenBookQA, and MedQA datasets, respectively. These results confirm that black-box knowledge injecting can effectively enhance the capabilities of LLMs.

- KnowGPT outperforms all KG-enhanced LMs significantly. This implies our black-box knowledge injection method proficiently encodes knowledge into LLMs. Furthermore, it showcases the superiority of our black-box approach, given its adaptable application to GPT-3.5 using only the model API, a feat not achievable by white-box methods.

### 4.2.1 Leaderboard Ranking

We submit our results onto the official leaderboard maintained by the authors of OpenbookQA. The full records on the leaderboard are shown on the website[2], while our result can be found from here[3].

We summarize the related submissions in Table 2, including three categories: traditional KG-enhanced LM, fine-tuning of LLMs, e.g., T5-11B used in UnifiedQA, and ensemble of multiple predictors. KnowGPT significantly outperforms traditional KG-enhanced LMs with 5.2% improvements when compared to the best baseline. The third group of methods occupies the leaderboard by leveraging ensemble learning strategies. Nevertheless, KnowGPT can still obtain competitive performance without ensembling with 0.6% above GenMC Ensemble [33]. Notably, our KnowGPT is remarkably comparable to the human performance.

---

[2]https://leaderboard.allenai.org/open_book_qa/submissions/public.
[3]https://leaderboard.allenai.org/open_book_qa/submission/cp743buq4uo7qe4e9750.

Table 2: OpenBookQA Official Leaderboard records of three groups of related models.

| **Human Performance (0.917)** | | | |
|---|---|---|---|
| w/o KG | 0.778 | UnifiedQA [36] | 0.872 |
| MHGRN [20] | 0.806 | DRAGON [79] | 0.878 |
| QA-GNN [80] | 0.828 | GenMC [33] | 0.898 |
| GreaseLM [88] | 0.848 | Human Performance | 0.917 |
| HamQA [15] | 0.850 | GenMC Ensemble [33] | 0.920 |
| JointLK [64] | 0.856 | X-Reasoner [32] | 0.952 |
| GSC [72] | 0.874 | KnowGPT | **0.926** |

## 4.3 Ablation Studies (RQ2)

To answer **RQ2**, we conduct two ablation studies. **First**, in Table 3, we measure the importance of the tailored reinforcement learning-based knowledge extraction module, i.e., $\mathcal{P}_{RL}$. Specifically, we compare it with the heuristic sub-graph extraction strategy, i.e., $\mathcal{P}_{sub}$. The performance is evaluated by directly feeding the extracted knowledge with the prompt format of 'Sentence', i.e., $\mathcal{F}_s$, to GPT-3.5. We also include 'w/o KG' as the baseline where GPT-3.5 is asked to independently answer the given question with no reasoning background provided. The results clearly indicate the vital role of our proposed knowledge extraction strategies. **Second**, we compare each of the three prompt formats subject to the same extracted knowledge. The detailed results are shown in Table 4. Though different formats perform similarly within the difference of 2.2% - 3.3%, they are particularly suitable for different kinds of questions. We illustrate this observation in the following case study section. Both ablation studies support the indispensability of each module, armed with a tailored deep reinforcement learning-based knowledge extraction and a context-aware prompt translation, our KnowGPT performs best on all three benchmark datasets.

Table 3: Ablation study on the effectiveness of two knowledge extraction methods.

| Knowledge Extraction | Model | CSQA | | OBQA | MedQA |
|---|---|---|---|---|---|
| | | IHdev | IHtest | Test | Test |
| w/o KG | GPT-3 | 0.539 | 0.520 | 0.482 | 0.289 |
| | GPT-3.5 | 0.735 | 0.710 | 0.598 | 0.487 |
| | GPT-4 | 0.776 | 0.786 | 0.910 | 0.763 |
| $\mathcal{P}_{sub}$ | GPT-3.5 | 0.750 | 0.739 | 0.865 | 0.695 |
| $\mathcal{P}_{RL}$ | GPT-3.5 | 0.815 | 0.800 | 0.889 | 0.755 |
| Ours | KnowGPT | **0.827** | **0.818** | **0.924** | **0.781** |

Table 4: Ablation study on different prompt formats for the extracted knowledge.

| Knowledge Extraction | Prompts | CSQA | | OBQA | MedQA |
|---|---|---|---|---|---|
| | | IHdev | IHtest | Test | Test |
| $\mathcal{P}_{sub}$ | $\mathcal{F}_t$ | 0.728 | 0.701 | 0.832 | 0.589 |
| | $\mathcal{F}_s$ | 0.750 | 0.739 | 0.865 | 0.695 |
| | $\mathcal{F}_g$ | 0.737 | 0.715 | 0.871 | 0.680 |
| $\mathcal{P}_{RL}$ | $\mathcal{F}_t$ | 0.782 | 0.769 | 0.853 | 0.739 |
| | $\mathcal{F}_s$ | 0.815 | 0.800 | 0.889 | 0.755 |
| | $\mathcal{F}_g$ | 0.806 | 0.793 | 0.906 | 0.762 |
| Full KnowGPT | | **0.827** | **0.818** | **0.924** | **0.781** |

## 4.4 Case Studies (RQ3)

For **RQ3**, we provide insights into how KnowGPT facilitates the prompt translation with a real case from CommonsenseQA. We visualize both the extracted knowledge and the textual inputs to GPT-3.5 in Figure 3. In this example, given the same extracted knowledge, GPT-3.5 answers correctly based on the sentence format that we provide. In contrast, it fails to answer the question with triples and graph descriptions. They clearly indicate the superiority of KnowGPT in an automatic context-aware prompt translation. We make the following observations: $(i)$ Triple format $\mathcal{F}_t$ is intuitively suitable for all the simple questions by directly indicating the one-hop knowledge. $(ii)$ Graph description may inevitably introduce noise to ensure the completeness and contextual fluency of the directed graph. In this example, since 'vacation' appears in both question and answer choices, over-emphasizing and connecting the knowledge about 'vacation' with other concepts in the graph misleads the model to make a prediction with an oblique focus. $(iii)$ Our KnowGPT has shown superior performance in automatically constructing suitable prompts for particular questions.

## 5 Limitation

Through our exploration, we realize the natural limitations of KnowGPT brought by real-world KGs. Existing KGs are automatically constructed based on online corpora. This inevitably introduces a considerable number of noisy triples into KGs. The noisy knowledge may mislead the LLMs

to wrong predictions despite the effectiveness of our prompting methods. In the future, we would leverage KnowGPT with off-the-shelf KG refinement algorithms to improve the quality of KGs.

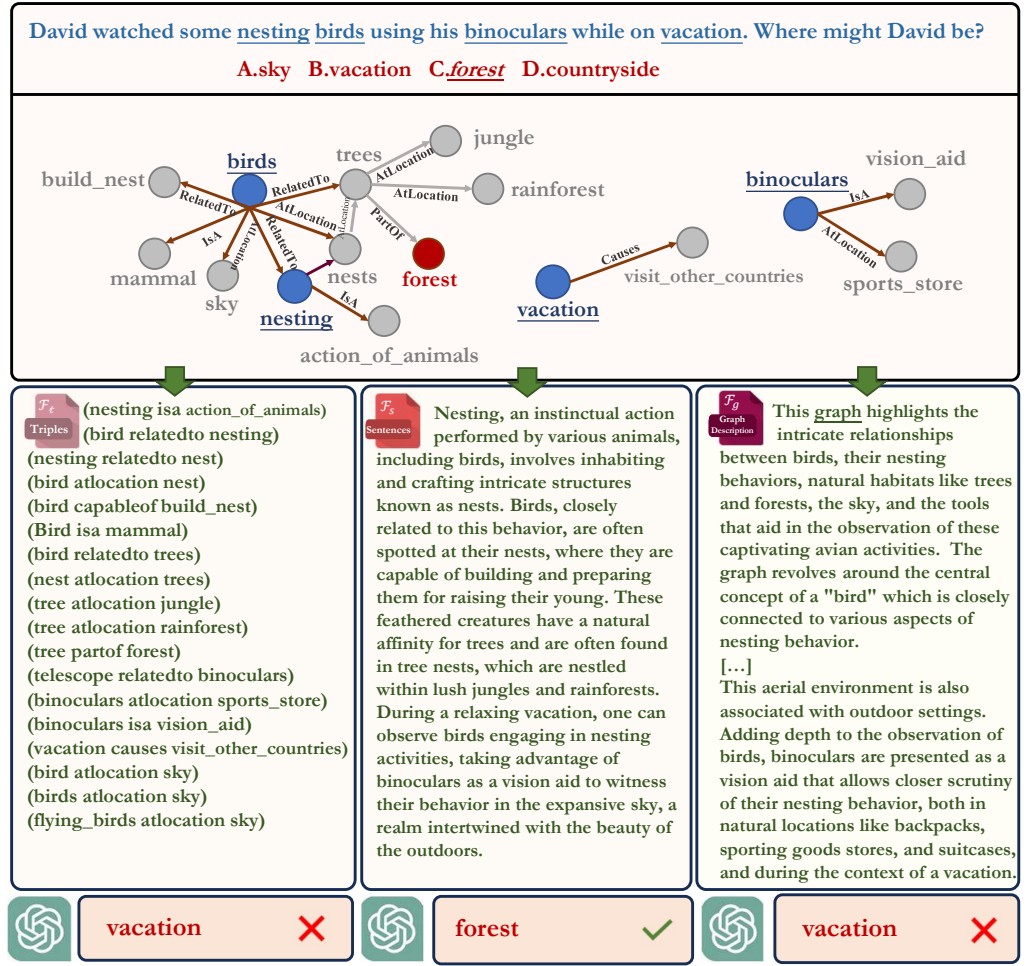

Figure 3: A case study on exploring the effectiveness of different prompt formats for particular questions. The extracted knowledge is shown in the middle of this figure in the form of a graph, where the nodes in blue are the key topic entities and the red is the target answer. The text boxes at the bottom are the final prompts generated based on three different formats.

## 6  Conclusion

The main objective of this paper is to tackle the hallucination issue that arises when applying LLM to specific domains. Although LLMs have strong reasoning capabilities, they still struggle to answer professional questions in specific domains, especially when the pre-training corpus lacks relevant knowledge. To address this issue, we propose a KG-augmented LLM model, named KnowGPT, which injects relevant domain knowledge from KGs into LLMs to assist the LLM in accurately answering professional questions. A novel framework, namely KnowGPT, is presented to integrate KGs into LLMs effectively with model APIs only. We first train a deep RL policy to extract informative and concise reasoning background from the KG. Then we learn an MAB to select the most effective knowledge extraction method and prompt template for each question. Extensive experiments on both general and domain-specific QA show superior performance of KnowGPT compared to all competitors.

## Acknowledgement

The work was fully supported by a grant from the Research Grants Council of the Hong Kong Special Administrative Region, China (Project No. PolyU 25208322).

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

# A   Related Work

To address the hallucination issues that arise when applying LLMs to specific domains, researchers have proposed various methods, which can be broadly categorized into three main categories: Fine-Tuning (FT), Retrieval-Augmented Generation (RAG) and KG-enhanced LLMs.

## A.1   Fine-tune LLMs with Domain-specific Data

To help the language model generate more accurate responses in specific domains, extensive studies have been conducted to investigate the efficacy of fine-tuning LLMs with specialized data [8, 27, 26, 39, 40, 22, 28]. Fine-tuning involves training the pre-trained LLM on a smaller dataset of domain-specific text, enabling the model to adapt its knowledge to the target domain, such as in recommendation [10, 91], and noGCNde classification [9, 30]. By exposing the LLM to domain-specific vocabulary, terminology, and patterns, fine-tuning enables the model to generate more accurate and relevant responses for domain-specific tasks [26, 31]. Fine-tuning LLMs with domain-specific data has been successfully applied across various domains. For example, in the healthcare domain, fine-tuned LLMs have been used for clinical note analysis [1], biomedical text mining [40], and medical dialogue [69]. Similarly, in the legal domain, fine-tuned LLMs have shown promise in tasks such as legal document classification [6], contract analysis [7], and legal judgment prediction [90].

However, recent research has highlighted the limitations and risks associated with fine-tuning LLMs using domain-specific data [37]. A study conducted by Google Research highlighted that using fine-tuning to update LLMs' knowledge can be problematic, particularly when the new knowledge conflicts with pre-existing information [23]. In such cases, acquiring new data through FT can lead to the model generating new hallucinations and even experiencing catastrophic forgetting [37, 78]. Besides that, many state-of-the-art LLMs are confined to a *black-box* role in practice. For instance, Calaude3 [3] and GPT-4 [54] exclusively grant access through their APIs, which means we can only retrieve model responses by submitting textual inputs, with model specifics inaccessible. This lack of access to model architecture and parameters prevents us from employing these fine-tuning methods.

## A.2   Retrieval-Augmented Generation for LLMs

Recently, Retrieval-Augmented Generation (RAG) models have been extensively explored to enhance LLMs with external knowledge from text corpora or online sources [41, 89, 53]. Combining LLMs with external knowledge retrieval systems can help reduce hallucinations. However, these approaches face challenges in domain-specific applications: $(i)$ Data quality. Domain knowledge is often scattered across different sources, such as textbooks, research papers, technical manuals, and industry reports [42]. These textual documents may have varying levels of quality, accuracy, and completeness, leading to potential inconsistencies or errors in the retrieved knowledge [92]. $(ii)$ Knowledge hierarchy. Domain knowledge is always complex and hierarchical and contains a high degree of jargon, acronyms, and specialized terminology. However, textual documents typically present this information in a linear and unstructured manner. The lack of explicit relationships and structured organization limits the reasoning and inference capabilities of RAG models, as they cannot leverage the structured connections within domain knowledge to derive new insights or generate more contextually appropriate responses [58]. $(iii)$ Huge search space. There is a lot of domain-irrelevant information in these textual documents, while domain-specific terminologies are always sparsely distributed over these documents [61]. The retrieval model can be computationally expensive and time-consuming, especially when dealing with large-scale knowledge sources, as the model needs to search through vast amounts of unstructured text to find relevant information [51].

## A.3   Integration of KGs and LLMs

KG-enhanced LLM aims to leverage the structured knowledge in knowledge graphs (KGs) [11, 48] to ground the model's responses in established facts and principles [29, 77, 56]. This grounding ensures that the generated outputs are based on reliable information rather than arbitrary or fabricated statements.

**Integrating KGs during Training**. Earlier studies adopted a heuristic way to inject knowledge from KGs into the LLMs during pre-training or fine-tuning. For example, ERNIE [63] incorporates entity

embeddings and aligns them with word embeddings in the pre-training phase, encouraging the model to better understand and reason over entities. KnowBERT [57] integrates entity linkers with BERT to introduce knowledge about entities during fine-tuning for downstream tasks.

**Integrating KGs during Inference**. Another line of work focuses on retrieving relevant knowledge from KGs at inference time to augment the language model's context. Typically, K-BERT [47] uses an attention mechanism to select relevant triples from a KG based on the input context, which are then appended to the input sequence. Similarly, KEPLER [73] learns joint embeddings of text and KG entities, enabling efficient retrieval of relevant knowledge to enhance the model's predictions. Later works further integrated graph neural networks alongside LLMs for joint reasoning [80, 87, 15] and introduced interactions between text tokens and KG entities within the intermediate layers of LLMs [64, 67].

**KG Prompting for LLM**. The methods above all assume they know everything about LLMs, including model architectures and parameters. However, as LLMs have been keeping evolving, many SOTA LLMs are confined to a *black-box* role in practice. As such, the research focus has shifted towards KG prompting that enhances fixed LLMs with KG-based hard prompts. KG Prompting for LLMs has been a new learning paradigm in natural language processing [46, 56]. Specifically, CoK [71] introduces a novel Chain-of-Knowledge prompting to decompose LLM-generated reasoning chains into evidence triples. It then verifies the triples' factuality and faithfulness using an external knowledge graph, ensuring the accuracy and reliability of the LLM outputs. By allowing the models to comprehend and reason over structured KG inputs, Mindmap [74] enhances the LLMs' ability to incorporate external knowledge, reduce hallucinations, and provide more transparency into their decision-making process. RoG [50] presents a planning-retrieval-reasoning framework that synergizes the strengths of LLMs and KGs to enhance the reasoning capabilities of language models in a more transparent and interpretable manner. Instead of retrieving factual information from KGs, KGR [25] proposes to autonomously retrofit the initial draft responses generated by LLMs. This innovative technique leverages the knowledge stored within KGs to mitigate hallucination by effectively extracting, verifying, and refining factual information throughout the entire reasoning process of LLMs.

**Our model**. Despite the promising performance of existing KG prompting methods, their extensive deployment and practical implementation in domain-specific applications are still impeded by two critical issues. ❶ The cost associated with calling the LLM API or deploying LLMs with cloud services is prohibitive. For example, GPT-4 is estimated to cost at least thousands of dollars for pilot-scale customer service, making the careful selection of the most informative triples from KGs essential to minimize costs. ❷ LLMs are highly sensitive to prompts, with even minor variations in prompts conveying the same semantic meaning potentially yielding drastically different responses. However, existing methods rely on manually designed or rule-based prompts to present factual knowledge from KGs. These hard prompts are always fixed and rigid, lacking the flexibility to adapt to variations in question semantics and KG structures. In this paper, we shed light on the whole community with a novel prompt learning method that is rather effective, generalizable and cost-efficient. There are two key components in our proposed KnowGPT including (i) knowledge retrieval, which leverages deep RL to extract relevant knowledge from KGs, and (ii) context-aware decision module to translate the structured knowledge from KGs into the appropriate prompt format.

# B    Implementation Details

## B.1    Entity Linking and Heuristic Knowledge Extraction

For each QA context, we adopt the methodology outlined in the prior research [44, 79] to extract the subgraph from the background knowledge graph (KG), denoted as $\mathcal{G}$. We commence by executing entity linking on $\mathcal{G}$, resulting in an initial collection of nodes, $V_{topic}$. Next, we incorporate bridge entities that appear within a 2-hop path between any two linked entities from $V_{topic}$, yielding the set $V_{retrieval}$. Subsequently, we refine this set by evaluating the relevance score for each node, adhering to the method proposed [79]. From this refined set, only the top 200 nodes, based on score, are retained. We then extract all edges connecting any pair of nodes in $V_{sub}$, creating the retrieved subgraph $G_{sub}$. Each node within $G_{sub}$ is designated a type based on its association to either the topic entities $Q$ or target entities $A$. Intuitively, the relevant reasoning background lies in a question-specific subgraph $\mathcal{G}_{sub}$ that contains all the *source* entities $S$, *target* entities $A$, and their

$k$-hop neighbors. Therefore, the reasoning background could be provided as the $\mathcal{G}_{sub}$, we denote this direct knowledge extraction method as $\mathcal{P}_{sub}$.

## B.2 Feature Initialization of Background KG

To calculate the initial node embeddings for entities extracted from the background KG, we adopt the approach proposed by the previous study [20]. Specifically, we transform knowledge triples from the KG into sentences and feed them into pre-trained LMs to get node embeddings. Specifically, to ensure a fair comparison, we implement all the KG-enhanced baselines and our model with the same advanced language models that are optimized for particular datasets. Specifically, RoBert-large [49] is used for CommenseQA, while AristoRoBERTa [13] is designated for OpenBookQA. For MedQA, we opt for the top-tier biomedical language model, SapBERT [45], to enhance comprehension of the biomedical field.

## B.3 Statistical Analysis of Datasets

We evaluate KnowGPT on three QA datasets spanning two fields: CommonsenseQA [66] and Open-BookQA [52] serve as benchmarks for commonsense reasoning, while MedQA-USMLE [34] acts as a domain-specific QA benchmark. While the official test set serves primarily for leaderboard rankings, we initially assess model efficacy using the in-house (IH) data split introduced in [44]. The official dataset is denoted as CSQA, while the IH split is represented by CSQA(IH)*. The statistics of these three datasets can be found in Table 5.

Table 5: The statistical information of three datasets.

| Dataset | Question | Choices | Train | Dev | Test |
|---------|----------|---------|-------|------|------|
| CSQA | #12102 | 5 | 9741 | 1221 | 1140 |
| CSQA(IH) | #12102 | 5 | 8500 | 1221 | 1241 |
| OBQA | #5957 | 4 | 4957 | 500 | 500 |
| MedQA | #12723 | 4 | 10178 | 1272 | 1273 |

**CommonsenseQA** is a multiple-choice question-answering dataset, each question accompanied by five potential answers. Answering its 12,102 questions necessitates a foundation in commonsense knowledge. While the official test set serves primarily for leaderboard rankings, we initially assess model efficacy using the in-house (IH) data split introduced in [44]. The official dataset is denoted as CSQA, while the IH split is represented by CSQA(IH)*.

**OpenBookQA**, commonly abbreviated as *OBQA*, comprises 5,957 multiple-choice questions, each offering four possible answers. To successfully answer these questions, one must have a comprehensive understanding of fundamental scientific facts and its applications.

**MedQA-USMLE**, abbreviated as *MedQA*, is a dataset consisting of 4-option multiple-choice questions that demand a grasp of biomedical and clinical understanding. These questions are sourced from preparatory tests for the United States Medical Licensing Examinations, and the dataset encompasses 12,723 questions. We adhere to the original data divisions as outlined in [34].

**Background Knowledge** To facilitate common sense reasoning, we employ ConceptNet [62], an extensive commonsense knowledge graph comprising more than 8 million interconnected entities through 34 concise relationships. For tasks specific to the medical domain, we leverage USMLE [80] as our foundational knowledge source. USMLE is a biomedical knowledge graph that amalgamates the Disease Database segment of the Unified Medical Language System (UMLS) [4] and DrugBank [75]. This repository encompasses 9,958 nodes and 44,561 edges.

## B.4 Implementation of Baselines

To verify the effectiveness of our proposed KnowGPT, we carefully selected baseline models from three aspects to ensure a comprehensive evaluation, among which Bert-base, Bert-large [35], and RoBert-large [49] are picked for being representative fine-tune LM methods; MHGRN [20], QA-GNN [80], HamQA [15], JointLK [64], GreaseLM [88] and GrapeQA [67] represent the state-of-art KG-enhanced LMs; ChatGLM [19], ChatGLM2 [81], Baichuan-7B, InternLM [68], GPT-3 [5],

GPT-3.5 [55] and GPT-4 [54] are picked for being representative generative large language models. To verify the effectiveness of our proposed prompting strategy used KnowGPT, we also add the state-of-the-art KG prompting methods, i.e., CoK [71], RoG [50], and Mindmap [74] as baselines. Notably, while some LLM baselines are actually open-source, we conduct the question-answering task under the zero-shot setting with the question query from the test set as input. All baseline methods used in this paper are based on their open-source implementations or officially-released APIs. Notably, we used the model 'text-davinci-002' provided by OpenAI as the implementation of GPT-3, and 'gpt-3.5-turbo' and 'gpt-4' as the implementations of GPT-3.5 and GPT-4, respectively. All models are implemented in Pytorch and trained on an RTX 3090 with 24 RAM. We use their best configuration as the default value for other distinctive hyperparameters in the baseline methods. To reduce the randomness, we use the random seed and report the average results of three runs.

## C Supplementary Results

### C.1 Compared to Fine-tune LLMs

To further verify the effectiveness of the knowledge injection framework, we also add several open-source trainable LLMs, i.e., ChatGLM, ChatGLM2, LLaMA-B, Baichuan-7B, InternLM and Vicuna-7B, and fine-tune them for commonsense reasoning on the benchmark CommonseQA. As shown in the Table 6, Our KnowGPT achieves comparable performance with no tuning on the LLM.

Table 6: Fine-tuned LLMs on three benchmark datasets.

| LLM | CommonsenseQA | OpenBookQA | MedQA |
|---|---|---|---|
| ChatGLM | 55.9% | 54.2% | 40.4% |
| ChatGLM2 | 60.0% | 55.6% | 45.5% |
| LLaMA-7B | 65.0% | 58.8% | 46.9% |
| Baichuan-7B | 58.8% | 56.1% | 43.4% |
| Alpaca-7B | 68.7% | 63.3% | 47.1% |
| Vicuna-7B | 66.7% | 64.0% | 45.2% |
| InternLM-7B | 75.2% | 70.7% | 48.3% |
| KnowGPT | **81.8%** | **92.4%** | **78.1%** |

### C.2 The effect of prompt format on different types of questions.

Generally, based on the complexity of the reasoning background, questions can be roughly categorized into three classes: (i) Simple question, like "What could be used as an electrical conductor?". (ii) Multi-hop reasoning question, like "Which school did Bill Gates' wife graduate from?" (iii) Global reasoning questions: "What do cats have in common with most mammals?"

Different types of questions correspond to different reasoning background. For example, simple questions only require basic factual triples, while multi-hop reasoning questions need a reasoning chain, and global reasoning questions require a more complex tree/graph-like reasoning background. To convert the extracted raw knowledge into textual prompt with corresponding logical reasoning structure, we designed three different prompt templates, including $F_t$, $F_s$ and $F_g$.

To verify the effectiveness of prompt formats, we conducted a statistical analysis on the benchmark datasets in terms of question types and then separately calculated the accuracy of different prompt formats on specific types of questions. As shown in Table 7, we can observe that triple-based prompt performs best on simple questions while graph description-based prompt performs better than any other prompt formats on complex questions. This is because graph description-based prompt could provide LLMs with more detailed and structured information by highlighting the local structure of the central entity.

In this part, we conduct comprehensive experiments to investigate the effect of prompt format on different types of questions. Table 7 presents the accuracy of different prompt formats on three types of questions. We observe that graph description-based prompt performs significantly better than any other prompt formats on complex questions. It is because graph description-based prompt could

Table 7: Accuracy of different prompt formats on specific types of questions on CommonsenseQA.

| Prompt Format | Simple | Multi-hop | Graph reasoning |
|:---:|:---:|:---:|:---:|
| $\mathcal{F}_t$ | **94.12%** | 74.47% | 42.10% |
| $\mathcal{F}_s$ | 88.23% | **82.98%** | 47.39% |
| $\mathcal{F}_g$ | 85.29% | 70.21% | **78.95%** |

provide LLMs with more detailed and structured information by highlighting the local structure of the central entity.

## C.3 The Effect of two Knowledge Retrieval Methods

If the background knowledge graph is complete and high-quality, our proposed RL-based knowledge extraction can ideally handle all cases. But in pratical implementation, we found that there are a few long-tail entities in KGs which only have few neighbors. In such cases, RL-based knowledge extraction can hardly retrieve reachable or effective paths due to the incompletion and sparsity of the background graph. Instead, directly extracting the whole subgraph through rule-based subgraph extraction can provide more knowledge for reasoning. The RL-based retrieval method $\mathcal{P}_{RL}$ could make sure our model extracts more concise and informative knowledge in most cases, while the rule-based subgraph extraction $\mathcal{P}_{RL}$ could supplement the former especially when the reasoning background is sparse. Thus, in this paper, we adopt Multi-Armed Bandit (MAB) to automatically select the most suitable retrieval methods from these two candidates, i.e., $\mathcal{P}_{RL}$ and $\mathcal{P}_{sub}$. The ablation study on the knowledge retrieval method can be found in Table 3 of the main content. Specifically, we compare our model with $\mathcal{P}_{RL}$, the heuristic sub-graph extraction $\mathcal{P}_{sub}$, and we also include 'w/o KG' as the baseline where the LLM is asked to independently answer the given question with no reasoning background provided. The results shown in Table 3 of our submission clearly indicate the vital role of our proposed knowledge extraction strategies.

## C.4 Efficiency Analysis: API Cost and Model Efficiency

To investigate the efficiency of our model, we compare it on MedQA with six representative baselines from three different categories, including traditional KG-enhanced LMs, i.e., JointLK and GrapQA, zero-shot LLMs, i.e., ChatGPT and GPT-4, and the SOTA KG-prompting-based LLMs, i.e., CoK and Mindmap. Notably, zero-shot LLMs, like ChatGPT and GPT-4, do not require training while traditional KG-enhanced LM methods like JointLK and GrapeQA are based on lightweight LMs and do not require interaction with LLMs, thus they do not incur API cost.

From Table 8, we can see that (i) Traditional KG-enhanced LM methods, like JointLK and GrapQA, have the shortest inference time. This is because the other models need to send requests to the Black-box LLM via API, and the extra response time of the LLM leads to long inference times for these models. Despite the efficiency, they have the worst performance. (ii) Our model outperforms the other models with comparable training time and the most economical API cost compared to models in the same category, including CoK and Mindmap. That is because we are the only model that consider the conciseness of knowledge during retrieval and prompt design.

Table 8: API cost and model efficiency analysis on MedQA.

| Model | Training time | Inference time | Avg tokens | Cost($) | Performance |
|:---|:---:|:---:|:---:|:---:|:---:|
| JointLK | 4.02 h | 0.13 h | - | - | 40.3% |
| GrapeQA | 4.49 h | 0.15 h | - | - | 39.5% |
| ChatGPT | - | 0.24 h | 98 | 1.87 | 48.7% |
| GPT-4 | - | 0.21 h | 98 | 29.77 | 76.3% |
| CoK | 5.85 h | 0.44 h | 1129 | 21.54 | 72.2% |
| Mindmap | 4.93 h | 0.36 h | 761 | 14.52 | 75.1% |
| KnowGPT | 5.47 h | 0.33 h | 348 | 6.64 | 78.1% |

## C.5 Stability Analysis

Due to page limits, we carefully described how we designed the reward function to ensure the RL model captures effective and informative knowledge in the main content of our submission. Actually, we have also employed the following techniques to stabilize the training process.

**Gradient clipping for policy network**. The policy network is updated using gradient-based optimization to maximize the expected cumulative reward. To stabilize the training of the policy network, we applied gradient clipping [21] to prevent the gradients from becoming too large. This technique helps mitigate the problem of exploding gradients and ensures stable updates of the network parameters. To demonstrate the effectiveness of gradient clipping, we conducted additional experiments comparing the performance of our model with and without gradient clipping across five runs. The results are presented in the table below. As shown in Table 9, the model with gradient clipping consistently achieves better performance across all datasets, demonstrating the stability and robustness of our method.

**Exploration-exploitation balance for MAB.** In our designed MAB, we have introduced a penalty term, i.e., $\beta$, to encourage exploration on under-explored arms (i.e., prompt formats), otherwise, it will lead to a greedy selection based on $c \times \alpha_{(i)}$. Table 10 shows the performance comparison between with and without this term.

Table 9: Stability study on gradient clipping.

| Model Variant | CSQA | OBQA | MedQA |
|---|---|---|---|
| w. gradient clipping | $81.8 \pm 0.51$ | $92.4 \pm 0.69$ | $78.1 \pm 0.66$ |
| w/o gradient clipping | $78.9 \pm 1.92$ | $90.2 \pm 1.25$ | $75.5 \pm 2.01$ |

Table 10: Stability analysis on penalty term.

| Model Variant | CSQA | OBQA | MedQA |
|---|---|---|---|
| w. penalty term | $81.8 \pm 0.51$ | $92.4 \pm 0.69$ | $78.1 \pm 0.66$ |
| w/o penalty term | $79.5 \pm 1.63$ | $91.0 \pm 1.45$ | $76.4 \pm 1.89$ |

# D  Broader Impacts

We are dedicated to not only focusing on the specific NLP task, i.e., knowledge-based QA, but also inspiring and benefitting a wide range of NLP communities with a novel knowledge injection framework to fast adapt LLMs to specific domains. Specifically, we would like to declare the following two contributions for a wide range of NLP communities.

**Exploring Effective Knowledge Injection for LLMs**: Applying LLMs for various NLP-related tasks in different domains is gaining momentum. However, fine-tuning LLMs for domain knowledge-based QA is expensive. Our paper presents a remarkably strong framework that could (i) retrieve the reasoning background from domain knowledge graph and (ii) automatically find out the optimal prompt to inject the domain knowledge into the LLM.

**Providing Insights in Prompt Learning**: While hard prompt is still dominating prompt learning for the majority of LLM-based scenarios, we shed light on the whole community with a novel prompt learning method that is rather generalizable and effective. We propose an auto-selection algorithm that can determine the best prompt format for LLMs to maximally understand the reasoning background.

