# OpenReview forum: "KnowGPT: Knowledge Graph based Prompting for Large Language Models"
_NeurIPS.cc/2024/Conference — NeurIPS 2024 poster_

### Official Review · Reviewer_t5Eo · 2024-07-10

**Soundness:** 2
**Presentation:** 2
**Contribution:** 2
**Rating:** 5
**Confidence:** 4

**Summary:**

This work presents KnowGPT, a novel KG-based Prompting framework designed to integrate domain knowledge into LLMs effectively. KnowGPT addresses the key challenges of previous methods, such as large search spaces, high API costs, and labor-intensive prompt engineering. The framework includes a knowledge extraction module that identifies the most relevant information from KGs and a context-aware prompt construction module that automatically generates effective prompts from the extracted knowledge. Experiments demonstrate that KnowGPT significantly outperforms existing methods, achieving a notable 92.6% accuracy on the OpenbookQA leaderboard, approaching human-level performance.

**Strengths:**

1.The topic of this paper is LLMs+KG, and KGs are indeed needed in practice to correct the outputs ofLLMs.

2.The method proposed in this paper is interesting; it adopts a search approach to determine the type of graph structure that is fed into LLMs and the form in which the prompt is presented.

3.The results of the method presented in this paper surpass most methods, demonstrating good performance.

**Weaknesses:**

1.This paper lacks necessary details regarding the proposed method KnowGPT, such as: (a) how the reinforcement learning part on the knowledge graph is trained; (b) how to optimize the Knowledge Graph Agent together with the Prompt Construction with Multi-armed Bandit as proposed in Section 3.2; (c) besides the two-hop subgraph extraction, the paper does not provide implementation details for the proposed method; (d) since we know prompts can significantly affect the output quality of models, the paper does not provide specific examples of inputs given to LLMs, only a case from KG to text, lacking the complete prompt information given to LLMs. This causes confusion: for LLMs, is it a multi-class or multiple binary (as in the YES/NO from Figure 2) classification problem? With multiple binary classifications, there is the issue of handling multiple (or zero) options being YES—how should this be dealt with? (Situations with all NO or all YES responses may arise.)

2.I think that the overall contribution of this paper primarily lies in a type of search algorithm: searching for subgraph structures (subgraph structures formed by RL or standard 2-hop subgraphs), searching for prompt forms (direct triples, triples converted into sentences, graph descriptions). The RL algorithm is fundamentally similar to Deeppath[1], and the authors should further distinguish the differences between them. Additionally, the authors should also elaborate on how the search algorithm proposed in this article differs from [2].

[1]Xiong, Wenhan, Thien Hoang, and William Yang Wang. "Deeppath: A reinforcement learning method for knowledge graph reasoning." arXiv preprint arXiv:1707.06690 (2017).

[2] Dong, Junnan, et al. "Active ensemble learning for knowledge graph error detection." Proceedings of the sixteenth ACM international conference on web search and data mining. 2023.

**Questions:**

1.On line 185 of the 5-th page, where are the additional training details mentioned in the appendix? Policy gradient is merely an algorithm, but how is the Agent on the Knowledge Graph trained? How is the overall objective function defined? Given that the action space in reinforcement learning on Knowledge Graphs is huge, how have you addressed this issue? From a technical solution perspective, what are the differences between the method 3.2 Knowledge Extraction with Deep Reinforcement Learning proposed in this paper and [1]?

[1]Xiong, Wenhan, Thien Hoang, and William Yang Wang. "Deeppath: A reinforcement learning method for knowledge graph reasoning." arXiv preprint arXiv:1707.06690 (2017).

2.What is the difference between "Graph Description" and "Sentences" in the prompt? Is the extracted subgraph structure the same for both, with the difference being that the "Graph Description" contains additional noise, such as "The entity is strongly associated with significant individuals in the tech industry."? I think a more reasonable approach would be that the "sentence" component targets the path, whereas the "Graph Description" component pertains to the entire subgraph structure.

3.On line 212 of the 6-th page, what does OLS refer to? Does it mean Ordinary Least Squares? What does \delta on line 218 of the sixth page refer to, is it a constant?

4.From a technical solution perspective, what is the difference between the content of Section 3.2 and the Active Ensemble Policy Learning with MAB described in [2]?

[2]Dong, Junnan, et al. "Active ensemble learning for knowledge graph error detection." Proceedings of the sixteenth ACM international conference on web search and data mining. 2023.

**Limitations:**

Yes

---

> ### Author Rebuttal · Authors · 2024-08-07
>
> Dear Reviewer t5Eo,
>
> Thanks a lot for your detailed feedback. We really appreciate your time and effort in pointing out the potential concerns related to our paper, and also, thanks a lot for the opportunity to further clarify the motivation, contribution, and technical details of our framework.
>
> > ### **Regarding W1: Training and Implementation Details.**
>
> A: Thank you for your valuable feedback. We appreciate your perspective on the core ideas presented in our paper. And also, thank you for the valuable opportunity to clarify the details and contribution of our framework.
> ### **Two-stage training**
> The training of our model consists of two steps:
> - We fisrt train an RL-based knowledge extraction model based on question-answer pairs in the training set, obtaining a well-trained policy network for knowledge extraction. After this stage, given a specific question, we can extract knowledge from the KG using two methods: **RL-based policy network** and the rule-based **subgraph extraction** method defined in Section B.1 of the Appendix.
> - We can obtain two sets of raw knowledge for each question by applying these two knowledge extraction methods (RL-based policy network and rule-based subgraph extraction). To transform this structured raw knowledge into easily understandable text for LLM, we design three different prompt templates for knowledge transformation. To automatically select the most suitable combination to construct the final prompt for LLMs, we trained a MAB model based on the data in the training set.
>
> After that, given a new question in the test set, we can automatically get the most suitable prompt in the text format based on the well-trained RL-based policy network and MAB model as follows.
>
> > For path extraction:
> >
> - Dataset: We extract the topic and choice entities from the original QA dataset. For each path, the source is set as the topic entity and the labeled target is the corresponding choice entity.
> - Training: In Section C.6 of the Appendix, we elaborate on the details of using the policy gradient to train our policy.
> - Evaluation: The overall path extraction performance is evaluated based on the cumulative rewards that the policy eventually obtains.
>
> > For MAB:
> - Dataset: (1) We prepare the candidate paths based on two knowledge extraction methods for each arm (RL-based and subgraph). (2) We also prepare three candidate prompt templates for each path. In general, we have six combinations: 2 raw knowledge * 3 prompt templates. (3) An online LLM for the feedback, 1/0, to indicate whether the constructed prompt is effective.
> - Training: Given a specific question, we train the MAB model based on the MSE loss.
> - Evaluation: The evaluation of whether the MAB made an effective selection is based on the feedback from LLM during the testing phase. We use the overall accuracy as the evaluation metrics.
>
> > ### **Regarding W2 (Q1,Q4): Compared to Deeppath and AEKE**
>
> A: We appreciate the reviewer's insightful comments and the opportunity to clarify the distinctions between our work and the cited papers. While our approach does involve searching for subgraph structures and prompt forms, we believe our contributions extend beyond just a search algorithm. Nevertheless, we acknowledge the importance of distinguishing our work from previous research.
>
> * **Compared to Deeppath**:
>     * **The task is diferrent**. DeepPath focuses on extracting a path from KG given start and end nodes. In contrast, our work addresses knowledge-based retrieval, extracting question-related knowledge while considering both KG structure and question semantics.
>     * **Reward function**. DeepPath's reward function is primarily based on KG structure, considering accuracy, path efficiency, and diversity. Our RL model introduces a more comprehensive reward scheme that promotes reachability, context-relatedness, and conciseness of the extracted knowledge.
>     * **Traing strategy**. While DeepPath uses supervised policy learning, our approach employs Gradient Clipping to maximize the expected cumulative reward, as detailed in Section C.6 of the Appendix. This difference in training strategy allows our model to better adapt to the complexities of knowledge-based question answering.
>
> * **Compared to AEKE:**
> Our MAB approach differs from AEKE's significantly in training: we've introduced a penalty term, β, to encourage exploration of under-explored prompt formats. This modification allows for a more balanced exploration-exploitation trade-off, which is crucial in our context of selecting optimal prompt forms. We show the performance comparison between with and without this term hereunder.
>
> | Model Variant | CSQA | OBQA | MedQA |
> | --- | --- | --- | --- |
> | w. penalty term | 81.8 $\pm$ 0.51 | 92.4  $\pm$ 0.69 | 78.1 $\pm$ 0.66 |
> | w/o penalty term | 79.5 $\pm$ 1.63 | 91.0 $\pm$ 1.45 | 76.4 $\pm$ 1.89 |
>
>
> > ### **Regarding Q2: Prompt Design**
>
> A: Generally, based on the complexity of the reasoning background, questions can be roughly categorized into three classes. Different types of questions correspond to different reasoning background. For example, simple questions only require basic facts in the form of triples, while multi-hop reasoning questions need a reasoning chain, and global reasoning questions require a more complex tree/graph-like reasoning background. To this end, we specifically designed three different prompt templates, including $\mathcal{F}_t$, $\mathcal{F}_s$, and $\mathcal{F}_g$, gudiding the LLM to convert the extracted raw knowledge into textual prompt with corresponding logical reasoning strucure as shown in Section 3.2.1 and C.6.
>
> > ### **Regarding Q3: Definition of OLS and $\delta$**
>
> A: Sorry for the confusion. OLS is the ordinary least square loss. While $\delta$ is a constant, for any $\delta > 0$ with the probability at lease $(1-\delta)$, the reward expectation $E(.)$ is bounded by a confidence interval. We'll include the detailed description in our revision.

---

> ### Author Response · Authors · 2024-08-12
> **Detailed response to Reviewer t5Eo (1/4)**
>
> Dear Reviewer t5Eo,
>
> We really appreciate your recognition of our work, and also appreciate your time and effort in providing insightful suggestions that can help further polish our paper. To make it clear, we provide more detailed responses to your comments and suggestions as follows:
>
> > **[W1-a & Q1-a,b]** How the reinforcement learning part on the knowledge graph is trained? On line 185 of the 5-th page, where are the additional training details mentioned in the appendix? Policy gradient is merely an algorithm, but how is the Agent on the Knowledge Graph trained?
>
> Thanks for your question. We appreciate your interest in the training details of our model. In our submission, we focused on describing how we carefully designed a meaningful reward function to ensure that the RL model captures effective and informative knowledge, and **the training detail is introduced in Section C.6 of Appendix**.
>
> To be clear, the policy network is updated using gradient-based optimization to maximize the expected cumulative reward. To stabilize the training of the policy network, we applied gradient clipping [1] to prevent the gradients from becoming too large. This technique helps mitigate the problem of exploding gradients and ensures stable updates of the network parameters. To demonstrate the effectiveness of gradient clipping, we conducted additional experiments comparing the performance of our model with and without gradient clipping across five runs. The results are presented in Table 1 below (you can also find the results in Table 10 of our original submission).
>
> Table 1: Abliation study for gradient clipping.
> | Model Variant | CSQA | OBQA | MedQA |
> | --- | --- | --- | --- |
> | w. gradient clipping | 81.8 $\pm$ 0.51 | 92.4  $\pm$ 0.69 | 78.1 $\pm$ 0.66 |
> | w/o gradient clipping | 78.9 $\pm$ 1.92 | 90.2 $\pm$ 1.25 | 75.5 $\pm$ 2.01 |
>
> As shown, the model with gradient clipping consistently achieves better performance across all datasets, demonstrating the stability and robustness of our method.
>
>
> > **[W1-b]** How to optimize the Knowledge Graph Agent together with the Prompt Construction with Multi-armed Bandit as proposed in Section 3.2?
>
> Thanks for the insightful comments. Ideally, we would like to design a end-to-end model: Given a LLM and a domain knowledge graph, the RL model could search from the KG to find the most concise, relevant, and informative knowledge according to the feedback from the LLM. This knowledge is then provided to the LLM to help it accurately answer professional questions. However, training such an RL model is extremely costly. Since RL-based knowledge retrieval is a Markov Decision Process, the RL agent would need feedback from the LLM at each decision step during training, requiring frequent interactions with the LLM and incurring huge API costs.
>
> To reduce the API cost, in this paper, we proposed a two-stage model, splitting the goal into two steps.
> * Knowledge extraction, which aims to extract the most relevant and informative knowledge from the knowledge graph for the target question. We designed three rewards to ensure that the RL model extracts correct, concise, and effective background knowledge.
> * Prompt Construction. It designs three different prompt templates for different questions and convert the extracted raw knowledge into a form that the LLM can most easily understand based on the LLM's feedback.
>
> The two-stage model can greatly reduce the API cost because we only need to interact with the LLM in the second stage.
>
> Back to your concern, the training of our model consists of two steps:
>
> - We fisrt train an RL-based knowledge extraction model $P_{RL}$ based on question-answer pairs in the training set, obtaining a well-trained policy network for knowledge extraction. After this stage, given a specific question, we can extract knowledge from the KG using two methods: RL-based policy network $P_{RL}$ and the vanilla rule-based subgraph extraction $P_{sub}$ defined in Section B.1 of the Appendix.
> - We can obtain two sets of raw knowledge for each question by applying these two knowledge extraction methods ($P_{RL}$ and $P_{sub}$). To transform this structured raw knowledge into easily understandable text for LLM, we design three different prompt templates for knowledge transformation. Therefore, we have six combinations: two raw knowledge * three prompt templates. To automatically select the most suitable combination to construct the final prompt for LLMs, we trained a MAB model based on the data in the training set.
>
> After that, given a new question in the test set, we can automatically get the most suitable prompt in the text format based on the well-trained RL-based policy network and MAB model.

---

> ### Author Response · Authors · 2024-08-12
> **Detailed response to Reviewer t5Eo (2/4)**
>
> > **[W1-c]** As for the two-hop subgraph extraction, the paper does not provide implementation details for the proposed method.
>
> Thanks for your insightful feeadback. **We include the implementation details of the vanilla 2-hop subgraph extraction $P_{sub}$ in Section B.1 of the Appendix.** To make it more clear, we will first explain the rationales behind the design of two knowledge extraction methods before summarizing the implementation details of $P_{sub}$.
>
> **The rationale behind knowldge extraction methods.**
>
> If the background knowledge graph is complete and high-quality, our proposed RL-based knowledge extraction $P_{RL}$ can ideally handle all cases. But in practical implementation, we found that there are a few long-tail entities in KGs which only have few neighbors. In such cases, RL-based path extraction can hardly retrieve reachable or effective paths due to the incompletion and sparsity of the background graph. Instead, directly extracting the whole subgraph through rule-based subgraph extraction can provide more effective knowledge. RL-based retrieval method $P_{RL}$ could make sure our model extract more concise and informative knowledge in most case, while the rule-based subgraph extraction $P_{RL}$ could supplement the former especially when the reasoning background is sparse.
>
> Thus, in this paper, we adopt Multi-Armed Bandit (MAB) to automatically select the most suitable retrieval methods from these two candidates, i.e., $P_{RL}$ and $P_{sub}$.
>
> **Experimental analysis of knowldge extraction methods.**
>
> The ablation study on knowledge retrieval method can be found in Table 3 of our submission. Specifically, we compare our model with  $P_{RL}$, the heuristic sub-graph extraction $P_{sub}$, and we also include ‘w/o KG’ as the baseline where the LLM is asked to independently answer the given question with no reasoning background provided. The results shown in Table 3 of our submission clearly indicate the vital role of our proposed path extraction strategies.
>
> **Implementation details of $P_{sub}$.**
>
> Due to page limits, we include the implementation details of the vanilla 2-hop subgraph extraction $P_{sub}$ in Section B.1 of the Appendix. Specifcally, we commence by executing entity linking on $\mathcal{G}$, resulting in an initial collection of nodes, $V_{topic}$. Next, we incorporate bridge entities that appear within a 2-hop path between any two linked entities from $V_{topic}$, yielding the set $V_{retrieval}$. After that, we refine this set by evaluating the relevance score for each node, following the previous study[1]. From this refined set, only the top 200 nodes, based on score, are retained. We extract all edges connecting any pair of nodes in $V_{sub}$, creating the retrieved subgraph $G_{sub}$. Each node within $G_{sub}$ is designated a type based on its association to either the topic entities $Q$ or target entities $A$.
>
>
> > **[W1-d]** For LLMs, is it a multi-class or multiple binary (as in the YES/NO from Figure 2) classification problem? With multiple binary classifications, there is the issue of handling multiple (or zero) options being YES—how should this be dealt with? (Situations with all NO or all YES responses may arise.)
>
> Thank you for making this valuable question. In this paer, we evaluate KnowGPT on three multi-choice QA datasets spanning two fields: CSQA and OBQA serve as benchmarks for commonsense reasoning, while MedQA acts as a domain-specific QA benchmark. Given the quesion with several choices, we let LLM give the specifc answer, like A/B/C/D, by providing it with background knowledge about the source entities from the original question and the target entities from all candidate choices. This approach transforms the task into a decision-making problem since we provide the LLM with comprehensive background knowledge for all candidate choices simultaneously, allowing it to make informed decisions based on a holistic view of the available information.

---

> ### Author Response · Authors · 2024-08-12
> **Detailed response to Reviewer t5Eo (3/4)**
>
> > **[W2-a & Q1-c,d]** The RL algorithm is fundamentally similar to Deeppath[1], and the authors should further distinguish the differences between them. Given that the action space in reinforcement learning on Knowledge Graphs is huge, how have you addressed this issue? From a technical solution perspective, what are the differences between the method 3.2 Knowledge Extraction with Deep Reinforcement Learning proposed in this paper and Deeppath[1]?
>
> We appreciate the reviewer's insightful comments and the opportunity to clarify the distinctions between our work and the cited papers.
>
> **Differences compared to Deeppath**:
>
> * **Motivation**. DeepPath focuses on extracting a path from KG given start and end nodes. In contrast, our work addresses knowledge-based retrieval, extracting question-related knowledge while considering both KG structure and question semantics.
> * **Reward function**. DeepPath's reward function is primarily based on KG structure, considering accuracy, path efficiency, and diversity. Our RL model introduces a more comprehensive reward scheme that promotes reachability, context-relatedness, and conciseness of the extracted knowledge. Especially, the **context-relatedness encourages paths closely related to the given question context, which can help the RL agent reduce the search space** since it can help filter out the irrelevant neightbors when considering the next step.
> * **Traing strategy**. While DeepPath uses supervised policy learning, our approach employs Gradient Clipping to maximize the expected cumulative reward, as detailed in Section C.6 of the Appendix. This difference in training strategy allows our model to better adapt to the complexities of knowledge-based question answering.
>
> > **[W2-b & Q4]** Additionally, the authors should also elaborate on how the search algorithm proposed in this article differs from AEKE[2]. From a technical solution perspective, what is the difference between the content of Section 3.2 and the Active Ensemble Policy Learning with MAB described in AEKE[2]?
>
> Thanks for the insightful comments. Our MAB approach differs from AEKE's significantly in training: we've introduced a penalty term, β, to encourage exploration of under-explored prompt formats. This modification allows for a more balanced exploration-exploitation trade-off, which is crucial in our context of selecting optimal prompt forms. We show the performance comparison between with and without this term hereunder.
>
> Table 2:  Ablation study on penalty term used in MAB-base prompt construction.
> | Model Variant | CSQA | OBQA | MedQA |
> | --- | --- | --- | --- |
> | w. penalty term | 81.8 $\pm$ 0.51 | 92.4  $\pm$ 0.69 | 78.1 $\pm$ 0.66 |
> | w/o penalty term | 79.5 $\pm$ 1.63 | 91.0 $\pm$ 1.45 | 76.4 $\pm$ 1.89 |
>
>
> > **[Q2]** What is the difference between "Graph Description" and "Sentences" in the prompt? Is the extracted subgraph structure the same for both, with the difference being that the "Graph Description" contains additional noise, such as "The entity is strongly associated with significant individuals in the tech industry."?
>
> Thanks for raising this point. Existing methods rely on manually designed or rule-based prompts to present factual knowledge from KGs. These hard prompts are inherently inflexible and rigid, lacking the adaptability to accommodate variations in question semantics and KG structures. To this end, we designed three different prompt templates, including $\mathcal{F}_t$, $\mathcal{F}_s$, and $\mathcal{F}_g$, gudiding the LLM to convert the extracted raw knowledge into textual prompt with corresponding logical reasoning strucure. An MAB is trained to automatically select the most effective prompt template for each question.
>
> **The rationale behind our prompt design**
>
> Generally, based on the complexity of the reasoning background, questions can be roughly categorized into three classes: (1) Simple question, like “What could be used as an electrical conductor?”. (2) Multi-hop reasoning question, like “Which school did Bill Gates' wife graduate from?” (3) Global reasoning questions: “What do cats have in common with most mammals?”
>
> Different types of questions correspond to different reasoning background. For example, simple questions only require basic factual triples, while multi-hop reasoning questions need a reasoning chain, and global reasoning questions require a more complex tree/graph-like reasoning background. To convert the extracted raw knowledge into textual prompt with corresponding logical reasoning structure, we designed three different prompt templates, including $\mathcal{F}_t$, $\mathcal{F}_s$, and $\mathcal{F}_g$.
>
> **Experimental analysis**
>
> Table 3. Accuracy of prompt templates on specific types of questions.
>
> | Prompt Template | Simple | Multi-hop | Global reasoning |
> | --- | --- | --- | --- |
> | $\mathcal{F}_t$ | 94.1% | 74.4% | 42.1% |
> | $\mathcal{F}_s$ | 88.2% | 82.9% | 47.3% |
> | $\mathcal{F}_g$ | 85.2% | 70.2% | 78.9% |

---

> ### Author Response · Authors · 2024-08-12
> **Detailed response to Reviewer t5Eo (4/4)**
>
> To verify the effectiveness of prompt formats, we conducted a statistical analysis on the benchmark datasets in terms of question types and then separately calculated the accuracy of different prompt formats on specific types of questions. As shown in Table 3, we can observe that triple-based prompt $\mathcal{F}_t$ performs best on simple questions, sentence-based prompt $\mathcal{F}_s$  work better for multi-hop questions  while graph description-based prompt $\mathcal{F}_g$ performs better than any other prompt formats on complex questions. This is because graph description-based prompt could provide LLMs with more detailed and structured information by highlighting the local structure of the central entity.
>
>
> > **[Q3]** On line 212 of the 6-th page, what does OLS refer to? Does it mean Ordinary Least Squares? What does $\delta$ on line 218 of the sixth page refer to, is it a constant?
>
> Thanks for your careful review. OLS is the ordinary least square loss. While $\delta$ is a constant, for any $\delta > 0$ with the probability at lease $(1-\delta)$, the reward expectation $E(r^{(i,b)} | {\bf x}_{i}^{(b)})$ is bounded by a confidence interval, as shown in h.
>
> **Thanks for your time and effort in providing insightful suggestions that can help further polish our paper. We're more than happy to provide any additional information or explanations that may be helpful in your review process.**

---

> > ### Comment · Reviewer_t5Eo · 2024-08-13
> >
> > Thanks for your responses. I have carefully read your responses, I think that my score is reasonable and fair. So I maintain the scores.

---

> ### Author Response · Authors · 2024-08-13
>
> Dear Reviewer t5Eo,
>
> Thank you for your recognition of our work and for providing such thorough and insightful feedback. We greatly appreciate the time and effort you've invested in reviewing our work and providing such detailed comments. Your comments and suggestions are invaluable in helping us improve the quality and clarity of our work.
>
> We have carefully considered each point you raised and have made substantial analysis and explanation. If there are any aspects that you feel still require further clarification or improvement, we would be more than happy to provide additional information or explanations to assist in your review process.
>
> Thank you again for your time and expertise. Your feedback has been instrumental in enhancing the quality of our research.

---

### Official Review · Reviewer_vKLu · 2024-07-11

**Soundness:** 3
**Presentation:** 3
**Contribution:** 3
**Rating:** 6
**Confidence:** 4

**Summary:**

The paper proposes KnowGPT, a novel KG prompting enhanced LLM framework that leverages deep reinforcement learning (RL) to extract knowledge and Multi-Armed Bandit (MAB) to generate effective prompts for domain-specific queries. Empirical evidence on QA benchmarks shows KnowGPT’s superiority over other methods.

**Strengths:**

1. The paper introduces a novel method with RL to extract question-related knowledge.
2. The paper design a tailored prompt construction strategy based on MAB.
3. The method achieves remarkable performance in experiments.

**Weaknesses:**

1. The analysis of the efficiency of the method seems to be insufficient.
2. The Introduction part seems to be too long.

**Questions:**

1. The paper argues that a big challenge is the high cost of using the API of LLM, but the efficiency analysis does not show how much it costs to call the API in order to train the two modules of the method, especially the prompt construction module.
2. The article mentions that the shortcoming of the existing method is that it requires Laborious prompt design, but your method also manually designs the prompt and requires effort of training. In addition, are the prompts of the latter two templates generated by LLM? Is the cost of this step factored into the efficiency comparison?
3. Can you provide the proportional distribution of different knowledge extraction methods and prompt forms during testing?

**Limitations:**

The authors discussed the limitations of their method in the paper.

---

> ### Author Rebuttal · Authors · 2024-08-07
>
> Dear Reviewer vKLu,
>
> We really appreciate your recognition of our work, and thanks a lot for providing insightful suggestions that can help further polish our paper.
>
>
> > ### **Regarding W1 (Q1,Q2): Efficiency Analysis**
>
> A: Thank you for the constructive comments. To make it more clear, we will first introduce the training and implementation details of our model before the detailed efficiency analysis.
>
> ### **Model Training**
> Ideally, we would like to design a end-to-end RL model: Given a LLM and a domain knowledge graph, the RL model could search from the KG to find the most concise, relevant, and informative knowledge according to the feedback from the LLM. This knowledge is then provided to the LLM to help it accurately answer professional questions. However, training such an RL model is extremely costly. Since the RL agent would need feedback from the LLM at each decision step during training, requiring frequent interactions with the LLM and incurring huge API costs.
>
> To reduce the API cost, in this paper, we proposed a two-stage model. The first step aims to extract the most relevant and informative knowledge from the knowledge graph for the target question, and the second step is to design three different prompt templates for different questions and convert this raw knowledge into a form that the LLM can most easily understand based on the LLM's feedback. **The two-stage model can greatly reduce the API cost because we only need to interact with the LLM in the second stage (prompt design).**
>
> ### **Efficiency analysis**
> Back to your concern, to investigate the efficiency of our model, we compare it on MedQA with 4 representative baselines from two different categories, including traditional KG-enhanced LMs and the SOTA KG-prompting-based LLMs, i.e., CoK and Mindmap.
>
> From Table 1, we can see that:
> * Traditional KG-enhanced LM methods, like JointLK and GrapQA,  have the shortest inference time. This is because the other models need to send requests to the LLM via API, and the extra response time of the LLM leads to long inference times for these models. Despite the efficiency, they have the worst performance.
> * Our model outperforms the other models with comparable training time and the most economical API cost compared to models in the same category, including CoK and Mindmap. That is because we are the only model that consider the conciseness of knowledge during retrieval and prompt design.
>
> Table 1. Efficiency analysis on MedQA.
> | Model | Training time | Inference time|Cost($)|Performance|
> | -------- | -------- | -------- | -------- | -------- |
> | JointLK     |4.02 h| 0.13 h| NA| 40.3%|
> |GrapeQA |4.49 h |0.15 h |NA |39.5%|
> |CoK |5.85 h |0.44 h| 21.54| 72.2% |
> |Mindmap| 4.93 h |0.36 h| 14.52 |75.1%|
> |KnowGPT |5.47 h| 0.33 h|6.64 |78.1%|
>
>
> > ### **Regarding W2: Overlong Introduction**
>
> A: Thanks a lot for the valuable suggestion. We’ll condense the introduction section by highlighting the key idea of our model.
>
> > ### **Regarding Q2: Laborious Prompt Design**
>
> A: Existing methods rely on manually designed or rule-based prompts to present factual knowledge from KGs.  These hard prompts are inherently inflexible and rigid, lacking the adaptability to accommodate variations in question semantics and KG structures. To this end, we specifically designed three different prompt templates, including $\mathcal{F}_t$, $\mathcal{F}_s$, and $\mathcal{F}_g$, gudiding the LLM to convert the extracted raw knowledge into textual prompt with corresponding logical reasoning strucure.  An MAB is trained to automatically select the most effective prompt template for each question.
>
> ### **The rationale behind our prompt design**
> Generally, based on the complexity of the reasoning background, questions can be roughly categorized into three classes as defined in Section C.3. Different types of questions correspond to different reasoning background. For example, simple questions only require basic factual triples, while multi-hop reasoning questions need a reasoning chain, and global reasoning questions require a more complex tree/graph-like reasoning background.
>
> Table 2. Accuracy of prompt templates on specific types of questions.
>
> | Prompt Template  | Simple  | Multi-hop  |Global reasoning|
> | -------- | -------- | -------- |-------- |
> |   $\mathcal{F}_t$   | 94.1%  | 74.4% |42.1%|
> |$\mathcal{F}_s$  | 88.2%| 82.9% |47.3%|
> |$\mathcal{F}_g$  | 85.2%| 70.2% |78.9%|
>
> As shown in Table 2, triple-based prompt $\mathcal{F}_t$ performs best on simple questions while graph description-based prompt $\mathcal{F}_g$ performs significantly better than any other prompt formats on complex questions. It is because graph description-based prompt could provide LLMs with more detailed and structured information by highlighting the local structure of the central entity.
>
> > ### **Regarding Q3: Proportional Distribution**
>
> A: Thank you for your valuable suggestion. We have conducted a thorough analysis on three benchmark datasets.
>
> Table 3. Proportional distribution of different knowledge extraction methods and prompt form on three datasets.
> | Dataset |CSQA|OBQA|MedQA|
> | ------------ | ------------ | ------------ | ------------ |
> |$P_{sub}$ : $P_{RL}$ (%)| 17.03 : 82.96|14.40 : 85.60|24.84 : 75.15|
> |$F_{t}$ : $F_{s}$ : $F_{g}$ (%)|52.33 : 32.35 : 15.31  |68.40 : 20.40 : 11.20|42.61 : 31.60 : 25.78 |
>
> From the table 3, we can see that:
> * Our RL-based knowledge extraction approach $P_{RL}$ was consistently favored across all datasets, which verifies the ability of model to find complex reasoning paths.
> *  The distribution of prompt forms varied more across datasets. The triple format were most prevalent in both CSQA and OBQA, likely due to there are lots of simple questions in these two datasets that the LLM could easily answer them with simplest knowledge format.
> *  Graph descriptions were used least frequently overall, but still played a significant role, especially in MedQA (25.78%).

---

> > ### Comment · Reviewer_vKLu · 2024-08-08
> >
> > Thank you for the detailed feedback and additional experimental results. I have updated the evaluation accordingly. Hope these analyses will be added to the final version. good luck!

---

> ### Author Response · Authors · 2024-08-08
> **Grateful Thanks to Reviewer vKLu**
>
> Dear Reviewer vKLu,
>
> We are deeply grateful for your strong support and for raising the confidence score. Your insightful suggestions are crucial in elevating the quality of our paper. We'll incorporate all the additional experiments and analysis in our final revision. Thank you for your time and expertise throughout this review process.

---

### Official Review · Reviewer_4L5K · 2024-07-13

**Soundness:** 3
**Presentation:** 3
**Contribution:** 3
**Rating:** 6
**Confidence:** 4

**Summary:**

The paper presents a novel framework, KnowGPT, for incorporating Knowledge Graphs into LLM-prompting-based question answering. It breaks down the problem into two main parts: identifying the concise, rich, and relevant subgraph for the question and using the Multi-Armed Bandits framework to select the best prompt construction method. The first part uses deep reinforcement learning with a three-part reward for training a model for subgraph identification. The second part uses the MAB framework to select the best subgraph representation and Prompt construction method. The presented framework is evaluated on three datasets and compared with a variety of baseline models. The results show that the proposed framework, KnowGPT, results in better performance than all of the baseline and is comparable to an ensemble model. The paper also presents an ablation study showcasing the impact of each of the two components.

**Strengths:**

S1.	The paper is very well written and discusses a very relevant problem given the increased usage of LLMs
S2.	The proposed method has been well defined with the rationale behind each term being presented along with the impact being evaluated
S3.	The authors also conduct extensive experiments to showcase the value of the framework.

**Weaknesses:**

W1.	The case studies are referenced in some places in the paper, indicating that they are present in the main paper and not in the appendix.

**Questions:**

Q1.	The method assumes that the source and target entities are given. However, a discussion of the impact of using an off-the-shelf model to extract them will be important to determine the significance of that assumption.

**Limitations:**

The authors have clearly outlined the limitations of the work.

---

> ### Author Rebuttal · Authors · 2024-08-07
>
> Dear reviewer 4L5K,
>
> Thanks a lot for your detailed feedback. We really appreciate your recognition of our work and also appreciate your time and effort in providing insightful suggestions that can help further polish our paper. Below are detailed responses to your comments and suggestions:
>
> > ### **Regarding W1: Move Case Study into Main Paper**
>
> Thank you for your valuable suggestion. Due to page limits, we included the efficiency analysis, case study, ablation study and some supplementary results in the Appendix. Following your advice, we’ll move the key contents, like the case study, into the main paper.
>
>
> > ### **Regarding Q1: Impact of Entity Extraction Model**
>
> Thanks for the insightful comments. In this paper, we focused on developing effective and efficient knowledge retrieval and injection methods, assuming entity extraction as a separate step to isolate the impact of our core contributions. Specifically, in the original paper, we assume that the source and target entities are given by using BERT as the default entity recognition model. Actually, discussing the impact of using an off-the-shelf model for entity extraction is valuable to speed up real-world applications of our method. We thank the reviewer for bringing this important point to our attention.
>
> To investigate the effect of entity recognition model, we conducted an additional experiment on the CSQA dataset to evaluate the impact of using off-the-shelf entity recognition models.
>
> Table 1. Comparisons of entity recognition model on CSQA.
> | Model |Bert|Roberta|XLnet|Llama2-7B|Llama3-8B|
> | -------- | -------- | -------- | -------- | -------- | -------- |
> |Precision|93.82%|94.31%|95.05%|95.51%|96.37%|
> |KnowGPT  |81.83%|-|-|82.30%|82.95%|
>
> Due to time limit, we only replace the BERT used in our model with the most advanced entity recognition model, Llama2-7B and Llama3-8B.  As shown in Table 1, we can see that most state-of-the-art entity recognition models achieve high accuracy (>90%) on this task. The performance of our method can be further improved by applying more advanced entity recognition models, like Llama2-7B and Llama3-8B.
>
> We thank the reviewer for bringing this important point to our attention. We will add these experiments and discussion in our revised version.

---

> > ### Comment · Reviewer_4L5K · 2024-08-09
> >
> > Thank you for your detailed responses to the questions and also to the other reviewers. I hope to see some of the analyses added to the final version of the paper. Given that there is still some work to do to strengthen the paper, I will retain my score of 6: Weak Accept.

---

> > > ### Author Response · Authors · 2024-08-09
> > > **Great Thanks to Reviewer 4L5K**
> > >
> > > Dear Reviewer 4L5K,
> > >
> > > Great thanks for your affirmation and valuable suggestions. We are delighted that you found our responses to the questions and other reviewers' comments are detailed and helpful. Following your suggestions, we'll incorporate all additional experiments and discussion in our final revision to further strengthen our work.
> > >
> > > In our previous response, due to time limits, we only replaced BERT in our model with two advanced entity recognition models. Since then, we have been keeping working on more comprehensive comparisons. Up to now, we have completed the comparison with 5 off-the-shelf models. As shown in Table 1, we list the entity recognition performance of these five models, their model size and the corresponding QA performance when applied to KnowGPT.
> > >
> > > Table 1. Comparisons of entity recognition model on CSQA.
> > > | Model |Bert|Roberta|XLnet|Llama2|Llama3|
> > > | -------- | -------- | -------- | -------- | -------- | -------- |
> > > |Model Size|110M | 125M| 117M|7B|8B|
> > > |Precision|93.82%|94.31%|95.05%|95.51%|96.37%|
> > > |KnowGPT  |81.83%|82.11%|82.27%|82.30%|82.95%|
> > >
> > > Thank you again for your expertise and recognition of our work. Your expertise and guidance have been invaluable in shaping our revisions. We commit to presenting you with a substantially improved and comprehensive manuscript by incorporating all additional experiments and analysis in our final revision.

---

### Author Rebuttal · Authors · 2024-08-07

In this paper, we propose a knowledge injection framework called KnowGPT, which injects knowledge from KGs into LLMs to assist the LLM in accurately answering domain-specific questions. Overall, we primarily focus on two research questions:
* Given a domain-specific question and a large-scale KG, how could we effectively and efficiently retrieve factual knowledge from KG that relevant to the question?
* To help LLM accurately answer domain-specific question, how could we transform the retrievaled knowledge into textual input(prompt) that is easily understandable for LLM?

To address these questions, we design KnowGPT, a two-stage model, including knowledge retrieval and prompt design. The training of our model consists of two steps:
* We fisrt train an RL-based knowledge extraction model based on question-choice pairs in the training set, obtaining a well-trained policy network for knowledge extraction. After this stage, given a specific question, we can extract knowledge from the KG using two methods: **RL-based policy network** and the rule-based **subgraph extraction** defined in Section B.1 of Appedix.
* Applying these two knowledge extraction methods (RL-based policy network and rule-based subgraph extraction), we can obtain two sets of raw knowledge for each question. To transform this structured raw knowledge into textual prompt that is easily understandable for LLM, we design three different prompt templates for knowledge transformation. Therefore, we have six combinations: 2 raw knowledge * 3 prompt templates. To automatically select the most suitable combination to construct the final prompt for LLMs, we trained a Multi-Armed Bandit (MAB) model based on the data in the training set.

After that, given a new question in test set, we can automatically get the most suitable prompt in the format of text based on the well-trained RL-based policy network and MAB model.

To make it more clear, we will introduce the details by answering the following questions.

**a. Why a two-stage framework instead of an end-to-end model?**

Ideally, we would like to design a end-to-end RL model: Given a LLM and a domain knowledge graph, the RL model could search from the KG to find the most concise, relevant, and informative knowledge according to the feedback from the LLM. This knowledge is then provided to the LLM to help it accurately answer professional questions. However, training such an RL model is extremely costly. Since the RL agent would need feedback from the LLM at each decision step during training, requiring frequent interactions with the LLM and incurring huge API costs.

To reduce the API cost, in this paper, we proposed a two-stage model. The first step aims to extract the most relevant and informative knowledge from the knowledge graph for the target question, and the second step is to design three different prompt templates for different questions and convert this raw knowledge into a form that the LLM can most easily understand based on the LLM's feedback.
**The two-stage model can greatly reduce the API cost because we only need to interact with the LLM in the second stage.**

**b. Why three types of prompt templates?**

Generally, based on the complexity of the reasoning background, questions can be roughly categorized into three classes:
* Simple question, like “What could be used as an electrical conductor?”.
* Multi-hop reasoning question, like “Which school did Bill Gates' wife graduate from?”
* Global reasoning questions: “What do cats have in common with most mammals?”

Different types of questions correspond to different reasoning background. For example, simple questions only require basic factual triples, while multi-hop reasoning questions need a reasoning chain, and global reasoning questions require a more complex tree/graph-like reasoning background. To convert the extracted raw knowledge into textual prompt with corresponding logical reasoning strucure, we specifically designed three different prompt templates, including $\mathcal{F}_t$, $\mathcal{F}_s$, and $\mathcal{F}_g$.
To verify the effectiveness of the prompt formats, we conduct a statistical analysis on the benchmark datasets in terms of question types and then separately calculated the accuracy of different prompt formats on specific types of questions. As shown in Table 2, we can observe that triple-based prompt $\mathcal{F}_t$ performs best on simple questions while graph description-based prompt $\mathcal{F}_g$ performs significantly better than any other prompt formats on complex questions. It is because graph description-based prompt could provide LLMs with more detailed and structured information by highlighting the local structure of the central entity.

Table 2. Accuracy of prompt templates on specific types of questions.

| Prompt Template  | Simple  | Multi-hop  |Global reasoning|
| -------- | -------- | -------- |-------- |
|   $\mathcal{F}_t$   | 94.1%  | 74.4% |42.1%|
|$\mathcal{F}_s$  | 88.2%| 82.9% |47.3%|
|$\mathcal{F}_g$  | 85.2%| 70.2% |78.9%|

**c. Why two knowledge retrieval methods?**

The RL-based retrieval method could make sure our model extract more concise and informative knowledge in most case, while the rule-based subgraph extraction could supplement the former, especially when the reasoning background is sparse. Thus, in this paper, we adopt MAB to automatically select the most suitable retrieval methods. The ablation study on knowledge retrieval method can be found in Table 3 of our submission. Specifically, we compare our model with  $P_{RL}$, the heuristic sub-graph extraction $P_{sub}$, and we also include ‘w/o KG’ as the baseline where the LLM is asked to independently answer the given question with no reasoning background provided. The results shown in Table 3 of our submission clearly indicate the vital role of our proposed path extraction strategies.

---

### Author Response · Authors · 2024-08-14
**General Response to Area Chairs and All Reviewers (1/2)**

Dear Area Chairs and Reviewers,

We sincerely thank you for your time, effort, and invaluable feedback during the author-reviewer discussion phase. Your insights have been crucial in helping us refine and improve our work. To facilitate the next stage of review and discussion, we would like to summarize the key contributions of our paper, the main points raised during the discussion and our response.

 ### **Motivation**

LLMs are often criticized for their tendency to produce hallucinations, wherein the models fabricate incorrect statements on tasks beyond their knowledge and perception. To alleviate this issue, we proposed a novel Knowledge Graph-based prompting framework, namely KnowGPT.  **KnowGPT aims to leverage the factual knowledge in knowledge graphs (KGs) to ground the LLM's responses in established facts and principles**.

 ### **Challenges**
Despite the promising performance of existing KG prompting methods, three critical issues hinder their widespread application in practice.
* **Huge Search Space.** Real-world KGs often consist of millions of triples, resulting in a vast search space when retrieving relevant knowledge for prompting.
* **High API Cost**. Closed-source LLMs, like GPT-4 and Claude 3, are accessible through proprietary APIs, which can incur significant costs when performing KG prompting at scale. Thus, careful selection of the most informative knowledge from KGs is essential to minimize costs.
* **Laborious Prompt Design**. Existing methods rely on manually designed or rule-based prompts to present factual knowledge from KGs.  These hard prompts are inherently inflexible and rigid, lacking the adaptability to accommodate variations in question semantics and KG structures.



 ### **Contribution**
KnowGPT contains a knowledge extraction module to extract the most informative knowledge from KGs, and a context-aware prompt construction module to automatically convert extracted knowledge into effective prompts.
* **Exploring Efficient and Effective Knowledge Extraction for KG Retrieval**. We introduced a novel RL-based retrieval method and designed three rewards to ensure that the RL model extracts correct, concise, and effective background knowledge.

* **Providing Insights in Prompt Learning**. While hard prompt is still dominating prompt learning for the majority of LLM-based scenarios, in this paper, we shed light to the whole community with a novel KG prompting method which is rather generalizable and effective. We propose an auto-selection algorithm which can determine the best prompt format for LLMs to maximally understand the given reasoning background.

* **Extensive Experiments**. To verify the effectiveness, **we compare KnowGPT with 22 baseline models including fine-tuning LMs, KG-enhanced LMs, prompt learning models and state-of-the-art LLMs augmented with KG prompting** on 3 public benchmarks and 7 fine-tuning LLMs on CSQA. We believe the results in Table 1 and 7 of our submission and the **`Top Rank on OpenBookQA Leaderboard`** can verify the effectiveness and generalizability of our model. Besides that, we also conducted sufficient ablation study, case study and efficiency analysis to study the effect of our model.

 ### **Main Discussion Points and Our Responses**
* **Impact of Entity Extraction Model**. In this paper, we focus on developing effective and efficient knowledge retrieval and injection methods, assuming entity recognition from the original question as a separate step to isolate the impact of our core contributions. Reviewer 4L5K suggested us to study the impact of entity extraction models. Following the suggestion, we conducted an additional experiment on CSQA by replacing BERT in our model with 4 advanced entity recognition models (Table 1 of the Rebuttal to Reviewer 4L5K). We believe that discussing the impact of using an off-the-shelf model for entity extraction is valuable to speed up real-world applications of our method.

---

> ### Author Response · Authors · 2024-08-14
> **General Response to Area Chairs and All Reviewers (2/2)**
>
> * **Efficiency Analysis**. Existing KG prompting methods often suffer from the huge API cost. To investigate the efficiency of our model, we compare it on MedQA with several SOTA baselines in terms of both API cost and training efficiency. The results can be found in Table 9 of our submission and Table 1 of the Rebuttal to Reviewer vKLu.
>
> * **Prompt Design**. The hard prompts used in existing KG prompting models are inherently inflexible and rigid, lacking the adaptability to accommodate variations in question semantics and KG structures. To this end, we specifically designed three different prompt templates, guiding the LLM to convert the extracted raw knowledge into textual prompt with corresponding logical structure. To make it clear, we first carefully explained the rationale behind our prompt design and then introduced the implementation details (the Rebuttal to Reviewer vKLu and t5Eo) following with a detailed experimental analysis (Table 1 of the Rebuttal to Reviewer vKLu).
>
> * **Proportional Distribution**. To verify the effect of the MAB-based prompt constriction strategy, we conducted a thorough analysis on three benchmarks (Table 3 of the Rebuttal to Reviewer vKLu) by investigating the proportional distribution of different knowledge extraction methods and prompt forms during testing.
>
> * **Implementation Details**. In our submission, we focused on describing how we carefully designed a meaningful reward function to ensure that the RL-based extraction model  $P_{RL}$ captures effective and informative knowledge, and the implementation details are included in the Appendix. Specifically, we introduced the details of sub-graph extraction strategy $P_{sub}$ in Section B.1, while the training details of RL model are included in Section C.6 of Appendix. To make it clear, we first explained the rationales behind the design of two knowledge extraction methods and then summarized the training details together with several supporting experiments (Table 1 of Official Comment to Reviewer t5Eo and Table 3 of our submission). We'll include a new Algorithm 1 in our revision to show the overall training process.
> * **Differences Compared to Deeppath and AEKE**. We clarified the differences between our work and the cited papers in both the Rebuttal and Official Comment to Reviewer t5Eo. Specifically, our model diverges from DeepPath in motivation, reward function and training strategy, while our MAB approach differs from AEKE's significantly in training since we introduced a penalty term to encourage exploration of under-explored prompt formats. To make it more clear, we included additional experiments to verify the effect of our tailored design (Table 1&2 of Official Comment to Reviewer t5Eo).
>
>
> Thank you again for your time and expertise throughout this review process. We deeply appreciate your expertise and recognition of our work. Each point raised has given us new perspectives to consider. We commit to addressing all the comments and incorporating the additional experiments and analysis in our final revision. Hope that this summary could facilitate the next stage of review and discussion.

---

### Decision · Program_Chairs · 2024-09-25

**Decision:**

Accept (poster)

**Comment:**

Summary (taken from t5Eo's excellent recap)
==============
This work presents KnowGPT, a novel KG-based Prompting framework designed to integrate domain knowledge into LLMs effectively. KnowGPT addresses the key challenges of previous methods, such as large search spaces, high API costs, and labor-intensive prompt engineering. The framework includes a knowledge extraction module that identifies the most relevant information from KGs and a context-aware prompt construction module that automatically generates effective prompts from the extracted knowledge. Experiments demonstrate that KnowGPT significantly outperforms existing methods, achieving a notable 92.6% accuracy on the OpenbookQA leaderboard, approaching human-level performance.

Metareview
===============

The reviews are all positive, but none are overwhelmingly so.  They identify no major technical issues, and only minor presentation issues, which I'm confident can be addressed in a revision.  Thus, I am recommending the paper for acceptance.

The argument for acceptance is that the use of bandit models to select how to prompt a model with a knowledge subgraph is technically interesting and novel (although I'd like to see an explicit comparison with DSPy, which has a similar ethos).  And this novelty pays off with good empirical results.

The weaknesses mentioned by the reviewers are relatively minor, and I infer that reviewers' relatively low scores were more a product of lack of excitement than serious flaws.  I agree with the lack of integration of the case studies and the lack of explicit details weaken the text of the paper.